# Implicit Multimodal Alignment: On the Generalization of Frozen LLMs to Multimodal Inputs

**Mustafa Shukor**[1] [*]          **Matthieu Cord**[1,2]

[1]Sorbonne University, [2]Valeo.ai

## Abstract

Large Language Models (LLMs) have demonstrated impressive performance on multimodal tasks, without any multimodal finetuning. They are the *de facto* building block for Large Multimodal Models (LMMs), yet, we still lack a proper understanding of their success. In this work, we expose frozen LLMs to image, video, audio and text inputs and analyse their internal representation aiming to understand their generalization beyond textual inputs. Our work provides the following **findings.** Perceptual tokens (1) are easily distinguishable from textual ones inside LLMs, with significantly different representations (*e.g.* live in different narrow cones), and complete translation to textual tokens does not exist. Yet, (2) both perceptual and textual tokens activate similar LLM weights. Despite being different, (3) perceptual tokens are implicitly aligned to textual tokens inside LLMs, we call this the implicit multimodal alignment effect (IMA), and argue that this is linked to architectural design, helping LLMs to generalize. This provide more evidence to believe that the generalization of LLMs to multimodal inputs is mainly due to their architecture. These findings lead to several **implications.** (1) We find a positive correlation between the implicit alignment score and the task performance, suggesting that this could act as a proxy metric for model evaluation and selection. (2) A negative correlation exists regarding hallucinations (*e.g.* describing non-existing objects in images), revealing that this problem is mainly due to misalignment between the internal perceptual and textual representations. (3) Perceptual tokens change slightly throughout the model, thus, we propose different approaches to skip computations (*e.g.* in FFN layers), and significantly reduce the inference cost. (4) Due to the slowly changing embeddings across layers, and the high overlap between textual and multimodal activated weights, we compress LLMs by keeping only 1 subnetwork (called $\alpha$-SubNet) that works well across a wide range of multimodal tasks. The code is available here: https://github.com/mshukor/ima-lmms.

## 1   Introduction

Large Language Models (LLMs) [1, 2, 3, 4, 5] represent a noteworthy advancement in recent AI developments. Building upon the success of LLMs, the next stride in this field involves extending beyond the textual modality, giving rise to Large Multimodal Models (LMMs) [6, 7, 8, 4]. A notable line of research involves connecting LLMs to visual encoders, while keeping them frozen and only training a connector with modest number of parameters [9, 10, 11, 12, 13, 14, 15, 16, 17]. These methods yield comparable performance [10] to large-scale multimodal models with significantly reduced computational and data budget.

Keeping all pretrained unimodal models frozen and only training couple of millions of parameters [9, 10, 12] is an interesting phenomenon to understand, with limited research trying to decipher it. To

---

[*]Contact: {firstname.lastname}@sorbonne-universite.fr

38th Conference on Neural Information Processing Systems (NeurIPS 2024).

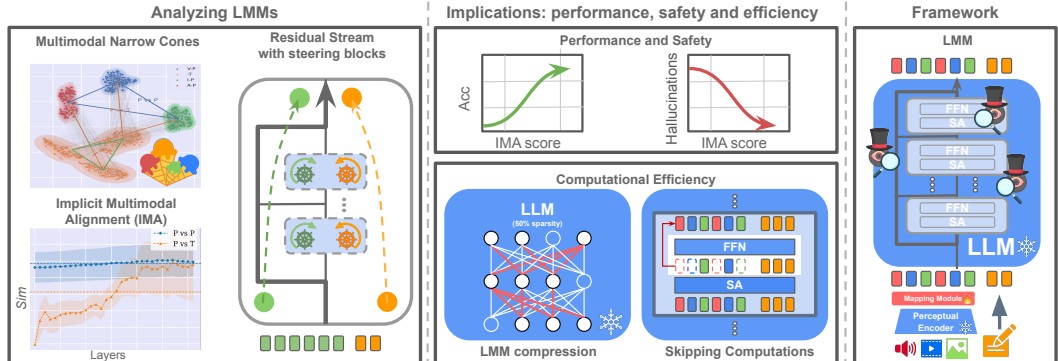

Figure 1: **Summary of the work.** We start by analysing multimodal tokens inside LLMs, and find that they live in different spaces (*e.g.*, multimodal cones). Yet they are implicitly aligned (*i.e.*, IMA), allowing us to see LLMs as residual streams with steering blocks. This lead to implications on performance, safety and efficiency.

explain why frozen LLMs can generalize beyond textual inputs, several hypotheses can be isolated: (1) perceptual tokens are transformed to textual ones, can be simply considered as foreign language [18], and thus the LLM sees text-only tokens. (2) LLMs are able to digest non-textual tokens that are processed by (2a) modality-specific subnetworks or (2b) the same LLM weights that can generalize due to other reasons.

In this study, we expose LLMs to different multimodal inputs, such as image, video, audio and text, and analyse their internal representations. We focus on frozen LLMs and consider two representative setups: single-task (ST) and multitask (MT) finetuning. The former is considered by parameter and data-efficient approaches [11, 12, 9, 10] and consists of training a mapping module for each dataset. The MT setup is considered by recent multimodal assistant models [19, 14, 20, 21], and consists of training the same mapping module on several datasets/tasks.

Our study shows (1) that perceptual and textual tokens still live in significantly different representation spaces inside LLMs (Sec. 3.1): live in different narrow cones, and have different norms, rate of change and vocabulary distributions. (2) We notice high similarity between the weights activated by textual and perceptual tokens (Sec. 3.2), allowing to swap these activated subnetworks between different tasks and modalities. Despite their differences, (3) perceptual tokens are implicitly aligned to textual ones across different stages (Sec. 4.1): during training of the mapping module, and during inference across LLM layers, especially inside each LLM block (*e.g.* after the self-attention layer). As there is no explicit objective to align these representations, we call it the Implicit Multimodal Alignment effect (IMA) (*i.e.*, increasing similarity between textual and perceptual token distributions). We find that this effect is mostly linked to architectural design (*e.g.* residual stream with refinement blocks acting as steering blocks Sec. 4.2). This provides more evidence to believe the architecture of LLMs is one of the main factors to generalize to multimodal representations.

We shed light on several practical implications (Sec. 5). (1) We find a positive correlation between the implicit multimodal alignment score and the task performance, suggesting that this score could act as a proxy metric. On the other hand, (2) we find a negative correlation with hallucinations, revealing that the main factor leading to this problem is the lack of alignment between the internal representation of textual and perceptual inputs. (3) The perceptual tokens slightly change inside LLMs, thus, we propose to skip their computations (*e.g.* inside FFN layers). (4) Due to the slowly changing embedding across layers, and the high overlap between weights activated by different modalities, we compress the LLM by keeping one task-agnostic subnetwork that works well across all modalities. To summarize (Fig. 1), we analyse the internal representations of LLMs when exposed to multimodal inputs, leading to the following **findings**:

- Perceptual and textual tokens live in different representation spaces inside LLMs.

- They activate similar LLM weights.

- They are implicitly aligned (IMA) inside LLMs, during training and during inference.

- The architectural design of LLMs can be perceived as a residual stream with steering blocks. We argue that this is one of the main factors allowing LLMs to: digest very different tokens, drive the implicit multimodal alignment effect, and thus generalize to different modalities.

These findings have several practical **implications** such as:

- The IMA score as a proxy metric candidate for task performance and hallucinations.
- Hallucinations as a result of lack of sufficient multimodal alignment.
- Skipping computations for visual tokens, leading to efficient inference.
- LLMs compression by keeping only 1 subnetwork that generalizes to all multimodal tasks.

## 2 Framework for analysing preceptually augmented LLMs

**General framework** We focus on a general family of models that consists of: a frozen language model $LLM$ with $L$ layers, a trainable mapping module $C$, and a frozen perceptual encoder $E_M$ for different modalities $M$ (*e.g.* image (I), video (V) and audio (A)). The $LLM$ input $X$ consists of the concatenation of $P = [p_1, ..., p_{N_p}]$ multimodal/perceptual tokens (referred to as prompt) with $T = [t_1, ..., t_{N_t}]$ textual tokens. The prompt $P$ is obtained after encoding the modality-specific input $XM$ with the corresponding $E_M$ and using $C$ to project it to the $LLM$ input space. $T$ is obtained from the embedding layer $E_T$ applied to the tokenized input text $XT$. This can be expressed as follows:

$$P = C(E_M(XM)), \quad T = E_T(XT), \quad O = LLM([P;T]). \tag{1}$$

The $k$ ($k = N_p + N_t$) output tokens $O = [o_i, ..., o_k]$ are obtained after a normalization, followed by the unembedding layer $W_{out}$ (or LLM head, *i.e.* $o_i = W_{out} LN_{out}(t_i^L)$). Our focus is on the internal representation of LLMs (*i.e.* tokens) at different stages, in particular across the $L$ LLM blocks/layers (referred to as B). The mechanism inside the $l + 1$ LLM transformer block can be expressed as follows:

$$X^{l+1} = X_{SA} + FC2(g(FC1(LN2(X_{SA})))), \qquad X_{SA} = X^l + SA(LN1(X^l)), \tag{2}$$

where $FC1$, $FC2$, $g$ are the up and down projections and activation inside the $FFN$, $LN1/2$ are the layer norms and $SA$ the self-attention.

**Perceptually augmented LLM baselines.** For the single-task (ST) setup, we train many models across different datasets that span image, video and audio-text modalities. Each mapping module is trained on a specific dataset, similar to previous works [10, 12, 9]. Inspired by previous studies [10, 12], we use light-weight transformer consisting of a self-attention to attend to perceptual tokens. In this setup, $P$ refer to perceptual tokens from image, video and audio modalities. For the multitask (MT) setup, we devise different variants of the LLaVA-1.5 [19] model that differ from the original model as follows: LLaVA-1.5-2 (LLM kept frozen), LLaVA-1.5-3 (LLM kept frozen, without pretraining) and LLaVA-1.5-4 (LLM kept frozen, without pretraining and with transformer mapping module similar to the ST setup instead of MLP). In this setup, $P$ refers to image tokens from different datasets. In the paper, we focus on LLaVA-1.5-4 as it is most similar to the ST setup, and analyse other variants in App. E. For analysis (*i.e.* Sec. 3), we focus on Vicuna-v1.5-7B [22] as it is shared by both setups. For the ST, we use unimodal encoders, such as ViT [23], TimeSformer [24] and AST [25] that are not aligned with text. More implementation details, and experiments with other backbones can be found in App. D and App. E. We report the similarity after averaging the tokens SimAvg (Eq. (3)) and details other measures in App. E.

**Analysis tools.** We are interested in cross-modal or multimodal alignment, and define the alignment in terms of the cosine similarity; the higher the score, the more the vector representations are pointing in similar directions. This could also indicates how much the two token distributions or vectors are close, in terms of L2 distance (assuming the vectors are normalized and in a narrow cones). In other words, alignment and similarity terms can be used interchangeably in the paper. In addition to cosine similarity we also study their norm, decoded vocabulary distributions and which LLM weights they activate. In the paper, we focus on the global representation per example, by analysing their average across the sequence. More finegrained analysis on the token level with different similarity and norm

measures gives similar observations and are detailed in App. E. For instance, we compute the cosine similarity between perceptual ($P$) (*e.g.*, tokens corresponding to image patches) and textual ($T$) tokens (*e.g.*, tokens corresponding to the image caption), after the block $l$ as follows:

$$\text{Sim}(P^l, T^l) = \frac{\hat{P}^l \cdot \hat{T}^l}{\|\hat{P}^l\| \|\hat{T}^l\|}, \quad \hat{P}^l = \frac{\sum_i^{N_p} p_i^l}{N_p}, \quad \hat{T}^l = \frac{\sum_i^{N_t} t_i^l}{N_t}, \tag{3}$$

## 3 LLMs indeed generalize to non-textual tokens

We investigate the generalization of LLMs to multimodal inputs, by studying the perceptual and textual tokens inside LLMs. We investigate if all tokens are projected to textual ones, or rather they are still different and how so (results with other models and similarity measures in App. E).

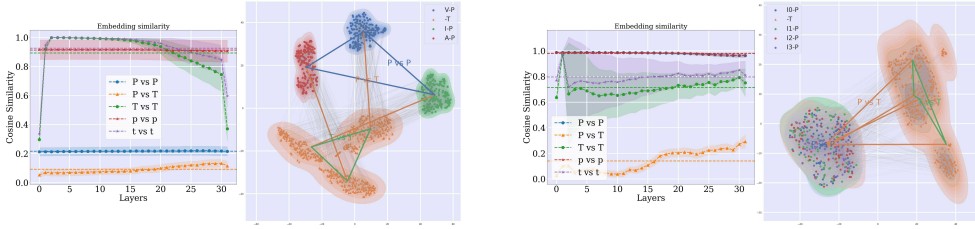

(a) ST setup. Tokens from multimodal datasets.     (b) MT setup. Tokens image-text datasets.

Figure 2: **Multimodal narrow cones.** The cosine similarity after LLM blocks (B) between: perceptual tokens ($P$ vs $P$), textual tokens ($T$ vs $T$), perceptual and textual tokens ($P$ vs $T$). $p$ vs $p$ and $t$ vs $t$ refer to the intra similarity within the same dataset. We also visualize the t-SNE of tokens (at layer 24) showing they stay separated inside the model. V (Video), I (Image), A (Audio).

### 3.1 How perceptual tokens differ from textual ones?

**Multimodal cones: different narrow cones for different modalities (Fig. 2).** Previous works [26, 27, 28, 29, 30, 31] have found the representation of contextualized embeddings in language models to be anisotropic: embeddings of different inputs exhibit high cosine similarity, shaping a narrow cone, where all embeddings point in the same narrow direction. In the multimodal domain, the cone effect is also observed [32] in contrastive models (CLIP [33]). In this section, we investigate if textual and multimodal tokens live in narrow cones inside LLMs, and if these cones are distinct. We compute the tokens cosine similarity at different layers. In particular, the unimodal similarity: text-only ($T$ vs $T$) and perceptual-only ($P$ vs $P$), and the cross-modal similarity ($P$ vs $T$) between perceptual and textual tokens. Note that for the ST setup, $P$ vs $P$ covers the similarity between image, video and audio tokens, while for the MT ones cover image tokens from different datasets. Fig. 2 shows a clear narrow cone effect for textual and perceptual tokens. Different perceptual modalities seem to live in different narrow cones, as shown by the low $P$ vs $P$ score for the ST setup. Interestingly, the cross-modal similarity between textual and perceptual tokens ($P$ vs $T$) is significantly lower, suggesting that textual and perceptual tokens also live in different narrow cones. We also visualize the t-SNE of the tokens embeddings showing they stay separated inside the LLM.

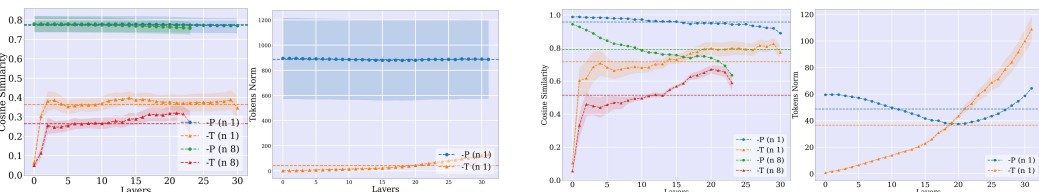

Figure 3: **Tokens norm and evolution across LLM layers.** The tokenwise cosine similarity between consecutive blocks (e.g. $X^{l+n}$ and $X^l$), and the median token L2 norm after each block ($X^l$) for the ST (left) and MT (right) setups. Textual and visual tokens evolve differently inside LLMs.

**Different token norms and evolution across layers (Fig. 3).** We compute the median of the token L2 norms after each LLM block. This shows that textual and perceptual tokens have different norms across layers. Perceptual tokens have significantly higher norm (at the beginning for MT and across all layers for ST), and they change significantly less. When looking at other norm measures, we found perceptual tokens with massive norms, similarly for textual ones [34], especially for the ST setup. We discuss massive tokens more in App. E.2. In addition, we compute the cosine similarity between tokens at block $l$ and block $l + n$, showing that textual and perceptual tokens have different change rates. Textual ones change drastically at the beginning of the LLM, while perceptual ones changes significantly less across all layers.

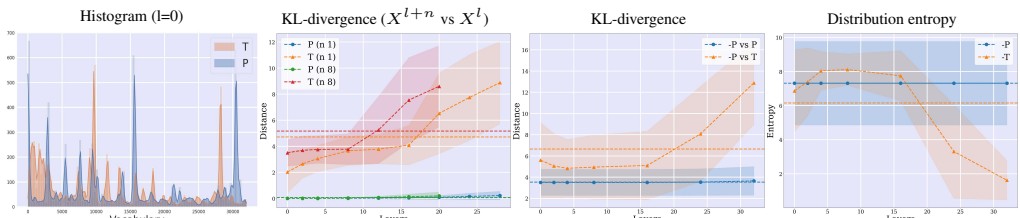

Figure 4: **Tokens vocabulary distribution inside LLMs.** The LLM (Vicuna-v1.5) unembedding layer is used to map each token at different LLM layer, to a probability distribution over the vocabulary. Multimodal tokens exhibit different vocabulary distributions across layers

**Different token vocabulary distributions across layers (Fig. 4).** For each token, we use the LLM unembedding (*i.e.* LLM head) to decode the latent representation to a probability distribution over the vocabulary. This approach have shown to work well for LLMs at different layers, not just the last one [35, 36, 37, 38]. We show the histogram of this distribution at the first LLM layer for both textual and perceptual tokens. The histograms show clear differences with some overlap between textual and perceptual prompts. In addition, we compute the KL-distance showing that the distributions diverge from each other across LLMs layers. We also notice that the distributions of textual tokens evolve significantly, compared to multimodal ones. This is shown by computing the KL-distance between consecutive blocks and the distribution entropy.

> *Finding* **1.** Textual and perceptual tokens live in significantly different representation spaces inside LLMs.

### 3.2 Do perceptual tokens traverse different paths inside LLMs?

For each trained model, we extract the LLM (frozen) subnetwork activated by each dataset/modality. We study these subnetworks (we refer to as pruning masks) by computing their similarity. We leverage the recent SoTA pruning approach (Wanda [39]), that prune models based on both the weights and the activation norms. Specifically, we use a handful (*e.g.* 256) of calibrated examples coming from different modalities, and keep only p% (1 - sparsity) of weights with the highest Wanda score, at different sparsity levels (30 % and 50 %). Note that after removing more than 50% of weights we observe a severe degradation of performance.

**Similar activated weights across modalities, in the first and deeper layers (Fig. 5).** Each subnetwork is represented as a binary mask to indicate which weights are activated. To compute the similarity between these networks, we consider the intersection over union (IoU). Results show an interesting high similarity between subnetworks activated by to different modalities. This high overlap is more seen for the ST setup, for example, the IoU between GQA and VQAv2 is 0.69, similar to GQA vs Audiocaps (0.67) or COCO-Text (0.65). When looking at the IoU across layers, we notice an interesting high score at first layers. It seems that first layers encode general features that are common for all modalities. This similarity increases as we go deeper in the LLM, moving to more abstract and less modality-specific representations, closer to the textual output.

**Transfer of multimodal subnetworks across tasks and modalities (Fig. 6).** To further validate the previous section, we study if we can simply interchange pruning masks between different tasks.

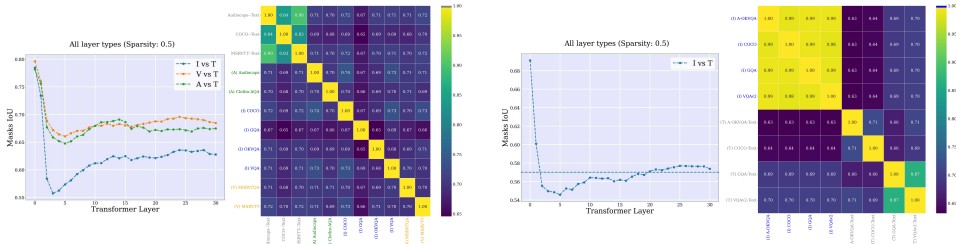

Figure 5: **IoUs of multimodal subnetworks**. IoU of the subnetworks activated by different tasks and modalities, for the ST (left) and MT (right) setups. We show the evolution of IoU across LLM layers and across different multimodal tasks. Different modalities activate similar LLM weights (Fig. 22 for clearer version of the figure).

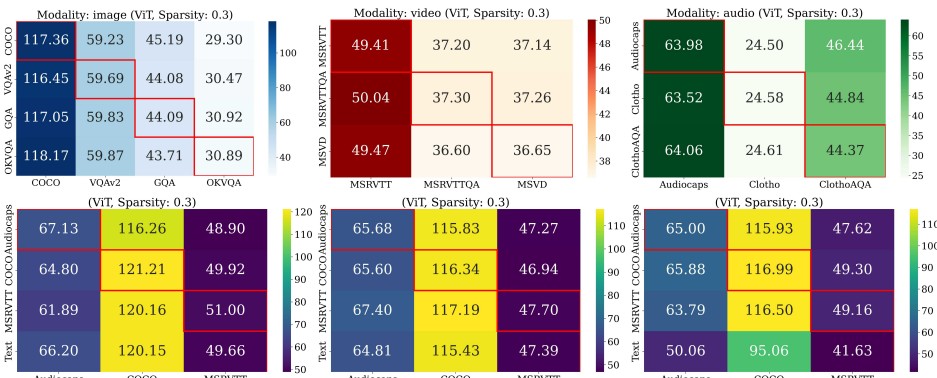

Figure 6: **Transfer of multimodal subnetworks across tasks and modalities**. The subnetwork activated by a given task is used for other tasks for Vicuna-v1.5. From left to right, transfer across: image, video and audio tasks. In each figure, the row corresponds to the subnetwork source dataset and the column to the target dataset. bottom: transfer across modalities for (from left to right): OPT, Llama 2, Vicuna-v1.5.

Specifically, for a given model trained to solve a particular task, we find the pruning mask using calibration data corresponding to different tasks/datasets. The sparsity is set to 30%, which is often used to maintain reasonable performance. Fig. 6 shows that the pruning masks transfer very well across tasks within the same modality (*e.g.* slight degradation by ∼1 point CIDEr for captioning with a mask coming from OKVQA). Similarly, we interchange masks across modalities. We fix the task (captioning) and also consider the text modalities (captions without images). In general, we observe similar transfer with a slight performance degradation, especially for OPT and Llama 2. We show similar observations with higher sparsity and with other encoders (*e.g.*, CLIP and MAE) in App. E.4.

**Modality-specific subnetworks?**  The experiments suggest a high overlap between weights activated by different modalities. However, this does not exclude the possibility of finding weights that are generally activated when seeing a particular modality, even if there are small amount of them. More discussion about this can be found in App. E.4.

> *Finding* 2.  LLM weights activated by perceptual and textual tokens overlap significantly.

## 4 What helps LLMs to generalize to multimodal tokens?

Textual and perceptual tokens have very different representations inside LLMs, yet, LLMs are still able to process and generalize to these non-textual tokens. In this section, we try to investigate why this is possible, in particular, we identify which factors facilitate this generalization.

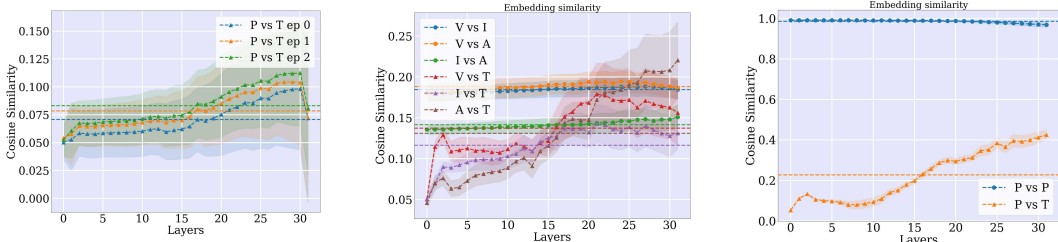

Figure 7: **Multimodal tokens similarity across LLM layers**. The cosine similarity between the textual and multimodal tokens across: training epochs i.e., 0, 1, 2 for Vicuna-v1.5 (first), and across LLMs layers: Vicuna-v1.5 (second) and LLaVA-1.5-4 (last). Textual and multimodal tokens are implicitly aligned during training, and during inference across LLM blocks.

## 4.1 Observation: the Implicit Multimodal Alignment Effect (IMA)

**Implicit alignment during training of the mapping module (Fig. 7).** We compute the cosine similarity between the perceptual tokens at the output of the mapping module, and textual tokens at different LLM blocks. Results show that this similarity increases at all layers. This reveals that the mapping module role, is not just to adapt the dimension of the visual tokens, but also to project the visual tokens to be semantically, as similar as possible to the textual ones.

**Implicit alignment during inference, across LLM blocks (Fig. 7).** We compute the cosine similarity between perceptual and textual tokens after each LLM block. Here we compute the max of tokenwise similarity (MaxSim App. E.1): for each pair of token sequences coming from one example (*e.g.* image prompt + caption), we take the maximum similarity, then we average across all examples. The tokenwise similarity between perceptual and textual tokens significantly increases, especially in the middle blocks, where the alignment is the highest in deep layers.

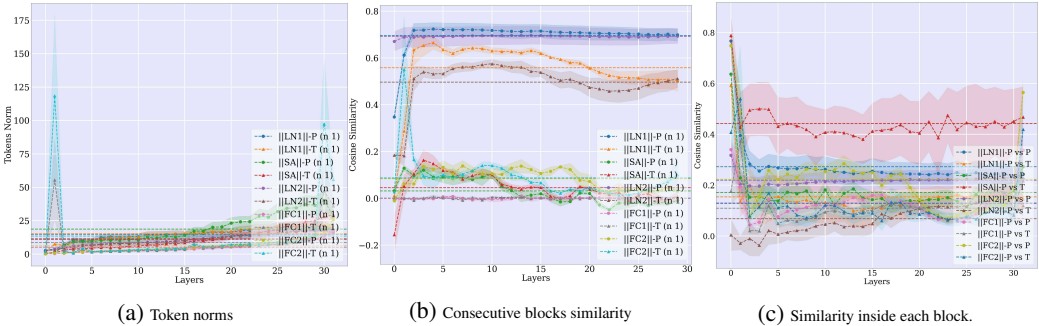

(a) Token norms      (b) Consecutive blocks similarity      (c) Similarity inside each block.

Figure 8: **Multimodal tokens norms and similarity inside LLM blocks.** Token norms (left), tokens cosine similarity between consecutive blocks (middle) and between perceptual and textual tokens (last). The tokens are inside Vicuna-v1.5 blocks (and outside the residual stream): after the self-attention (SA), and FFNs (FC1/2) and layer norms (LN1/2). Multimodal tokens are Implicit alignment inside LLM blocks.

**Implicit alignment during inference, inside LLM blocks (outside the residual stream) (Fig. 8).** We look deeper to investigate the source of this alignment, and focus on tokens inside the LLM block, which consists mainly of a self-attention (SA), FFN (FC1/2) and layer norms (LN1/2) layers. Interestingly, the similarity between textual and multimodal tokens is the highest after the SA layers.

> *Finding* **3.** An implicit multimodal alignment emerges to pull the textual and perceptual tokens closer inside LLMs, during training and inference.

## 4.2 Explanation: the architectural inductive bias hypothesis

**Residual stream with refinement blocks.** We notice different observations between the tokens inside and outside the residual stream. In the residual stream, the perceptual and textual tokens exhibit significant representation differences (Sec. 3.1), while outside the residual stream, they are more aligned. Each block contributes slightly to the residual stream (small token norms inside the blocks Fig. 8a), with significantly different contributions (cosine similarity between consecutive blocks close to zero, *e.g*, after the FC1/2 Fig. 8b). This allows us to view the model as a series of refinement blocks that try to gradually refine the input signals. As the original signals are significantly different, they stay different in the residual stream throughout the model. We argue that this provide a flexibility to handle too different inputs. Moreover, previous works [40] have shown that transformers contain both elements with high and low complexity biases, which helps to build general-purpose architectures [41] that are able to generalize. These works support further our findings.

**Refinement blocks as steering blocks.** Inside the transformer block, we notice that the layer normalization play an important role in having comparable norms for both textual and perceptual tokens (Fig. 8a). Perceptual token norms become smaller and closer to textual ones as we traverse several layers in the block. In terms of cross-modal similarity, we notice the highest similarity after the SA, then after the FC2 and LN1. Note that this similarity is higher inside the block, than in the residual stream (*e.g.* 0.45 vs 0.1 for Vicuna-v1.5 and 0.58 vs 0.15 for LLaVA-1.5-4 in the residual stream Fig. 2). After each block the cross-modal alignment increases, and hence the narrow cones are steered to each other. This suggests that all layers play an important role in steering the textual and perceptual narrows cones to be aligned, with the most contributions coming from the SA.

> *Finding* **4.** An LLM can be seen as a residual stream with refinement blocks acting as steering blocks. This architecture design plays an important role in generalizing to very different tokens, and hence other modalities.

## 5 Implications: performance, safety and efficiency

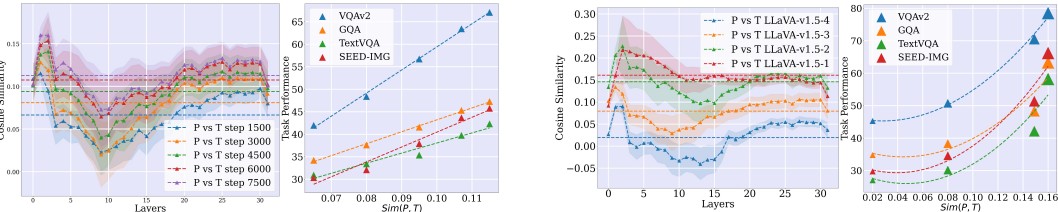

Figure 9: **Implicit alignment as a proxy metric for task performance.** Left: different checkpoints of LLaVA-1.5-4. Right: different variants of the LLaVA-1.5 model. We show the cross-modal token cosine similarity across layers, and the task performance across different benchmarks.

**Implicit alignment as a proxy metric for task performance? (Fig. 9)** We compute the cosine similarity between perceptual tokens at the LLM input and the textual tokens across LLM layers. The similarity increases during training. Interestingly, we notice a clear and positive correlation with the task performance on several multimodal benchmarks. In addition, we find that this correlation exists across different models, as shown for different LLaVA-1.5 variants.

**Implicit alignment as a proxy metric for hallucination? (Fig. 10)** Previous works have shown that LMMs suffer from severe hallucinations [42, 43, 44], and generally try to tackle this problem by training on better datasets [45], using RLHF or RLAIF [46, 47] or post-training heuristics [48, 49]. Here we highlight one of the main causes of hallucinations: which is the lack of internal alignment between textual and perceptual representations. we show the cosine similarity between textual and perceptual tokens after each LLM block, and report the hallucinations on POPE [50] and COCO [51] benchmarks. The curves show clear correlation between the implicit alignment and the hallucinations.

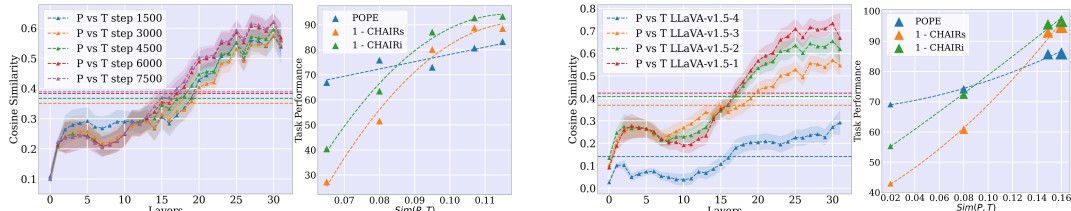

Figure 10: **Implicit alignment as a proxy metric for hallucinations.** Left: different checkpoints of LLaVA-1.5-4. Right: different variants of the LLaVA-1.5 model. We show the cross-modal token cosine similarity across layers, and the hallucinations across different benchmarks.

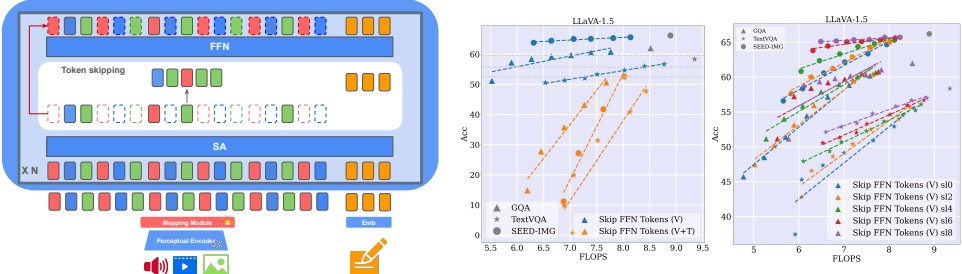

Figure 11: **Skipping computations for visual tokens**. Skipping (Skip ratio)% of the tokens in the FFN layers. sl: skipping start layer. (V): visual tokens. (T): textual tokens. Results on the MT (with LLaVA-1.5) setup.

**Skipping computations for visual tokens (Fig. 11).** In Sec. 3.1 we show that perceptual tokens change significantly less across layers, compared to textual ones. Sec. 4 highlights the importance of SA layers for cross-modal alignment. In this section, we leverage these observations to reduce the LLM computation overhead by skipping the computations of visual tokens. Specifically, starting from a given start layer (sl), we reduce computations in FFN layers, which accounts for almost 2/3 of model weights, by skipping p% (Skip ratio) of visual tokens. Fig. 11 shows that skipping the visual tokens leads only to slight decrease in performance, while reducing significantly the amount of compute. We provide additional results with the ST setup, and ablation study in App. F.3.

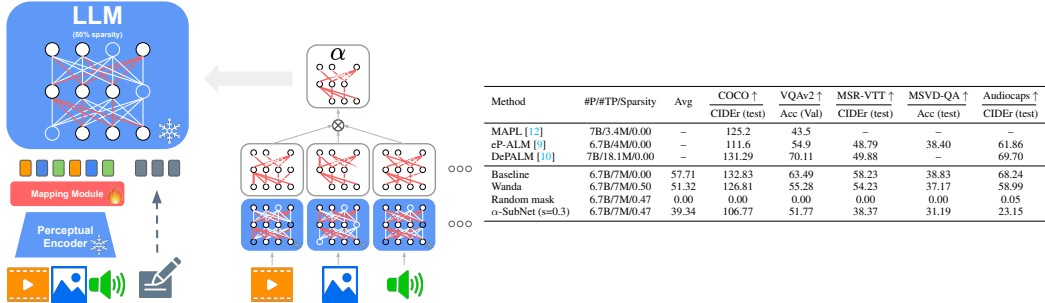

Figure 12: $\alpha$-**SubNet: a modality-agnostic subnetwork**. Left: illustration of how we obtain the $\alpha$-SubNet. Right: different methods to compress multimodal LLMs (OPT). Table 2.

$\alpha$-**SubNet: one LLM Subnetwork for all multimodal tasks (Fig. 12)** . Despite their differences, multimodal tokens share an important property: slowly changing embeddings across layers (Sec. 3.1). This suggests the possibility of compressing the model while retaining reasonable performance. In addition, textual and multimodal tokens are pulled closer inside the LLM (Sec. 3.2), and processed by almost the same LLM weights (Sec. 4), especially for the ST setup. This suggests the possibility of finding a common subnetwork ($\alpha$-SubNet) that works well for all multimodal tasks. Thus, we focus on the ST setup with the OPT and CLIP encoders that are currently used by previous works. We consider two representative tasks: COCO image captioning and VQAv2 and provide similar results for other tasks, and modalities in App. F.4. First we use Wanda for task-specific pruning

and show in Fig. 12 that we can obtain scores close to the original ones while removing 50% of the weights. To find the task and modality agnostic $\alpha$-SubNet, we first extract many pruning masks (*e.g.* at 30% sparsity) for different modalities, then take the intersection of all these masks (*e.g.* leading to a global mask at $\sim$ 50% sparsity). This approach is significantly better than other baselines such as magnitude pruning or a random mask, and leads to comparable performance compared to the task-specific Wanda pruning, especially for VQAv2.

## 6  Discussion

**Limitations.**   The paper focuses on open-source and frozen LLMs up to 7B parameters, LMMs that concatenate perceptual tokens at the LLM input and are relatively efficient. The generalization of our findings, to larger and more powerful models, with different architectures, including proprietary ones remains to be seen. Detailed discussion in App. B.

**Conclusion.**   We propose the first study of the internal representation of frozen LLMs when exposed to multimodal inputs. We find very different representations for perceptual and textual tokens, yet LLMs are still able to generalize to these non-textual tokens. The implicit multimodal alignment (IMA) effect, linked mostly to architectural design, facilitates this generalization by bringing multimodal tokens closer inside the LLM. Our findings have several implications, such as as reducing the computation resources at inference time, understanding better the performance as well as safety-related problems such as hallucinations. We hope that this study will have positive impact, pushing for more works to understand multimodal LLMs, and pave the way to devise more powerful models that are better aligned to human preferences, while targeting safety-related issues.

## 7  Acknowledgments

The authors would like to thank Arnaud Dapogny and Edouard Yvinec for fruitful discussions, and Damien Teney and Alexandre Ramé for their helpful feedback on the paper. This work was partly supported by ANR grant VISA DEEP (ANR-20-CHIA-0022), and HPC resources of IDRIS under the allocation 2024-[AD011013415R2] made by GENCI.

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

# Supplementary material

This supplementary material is organized as follows:

- App. A: detailed related work.
- App. B: detailed discussion about the work, limitations and other implications.
- App. C: the broader impact of the work.
- App. D: implementation details, including the trained models, datasets and metrics.
- App. E: additional experiments analysing LLMs.
- App. F: additional experiments for the implications.

# A  Detailed related work

**Large multimodal models.**  Motivated by the success of large-scale training of LLMs [52, 1, 3, 53, 2, 5, 4], the multimodal community has embarked on a parallel journey, striving to develop larger and more powerful models capable of processing multiple modalities. Typical Large Multimodal Models (LMMs) are constructed either by building upon frozen LLMs [6, 54, 55] or by training them end-to-end after initialization [56, 7, 8]. These models have demonstrated success in numerous general and intricate multimodal tasks, achieving performance levels close to human capability. Another important line of research focuses on unified models, where a single model is designed to handle diverse tokenized modalities, such as image-text [57, 58, 59], or even beyond two modalities [60, 61, 62].

**Efficient large multimodal models.**  Recently, to mitigate the training cost associated with training large multimodal models, efficient adaptation of unimodal models has emerged as a promising direction. Models like [11, 9, 17, 12, 10, 63] maintain LLMs frozen and train only a small subset of adaptation parameters for different multidmodal tasks. These approaches achieve competitive performance compared to end-to-end trained models [64, 65, 66, 67, 68] on image-text tasks and also on audio and video-text tasks [9, 69, 70, 10]. Beyond single-task tuning, many approaches do relatively light-weight pretraining and/or instruction tuning [71, 72, 14, 73, 19] and achieve good zero-shot generalizaton and instruction following abilities. To make these models more efficient, previous works have trained models with smaller LLMs showing competitive performance [74, 75, 76, 77, 78]

**Analyzing LLMs.**  Previous research has highlighted the highly anisotropic nature of embeddings within language models, characterized by high cosine similarity [26, 27, 28, 29, 30]. Studies focusing on efficiency have shown that textual tokens exhibit small changes across layers [79, 80, 81]. Additionally, LLMs contains outlier features [82] and massive activations [34], which significantly influence model performance. Work by [83] suggests that LLMs may generalize due to the fractal structure of language. Moreover, [40] demonstrate that the building blocks of LLMs implicitly bias towards approximating both complex and simple functions. In the multimodal domain, [32] identify a modality gap in CLIP models attributable to the narrow cone effect. Previous studies have also explored neurons in LLMs that encode multimodal representations [84, 85].

**Compression and pruning for multimodal models.**  Few works have targeted multimodal model compression, focusing mostly on image-text models. Notably, some works have concentrated on distilling knowledge from larger models through attention maps [86] or the affinity matrix in CLIP models [87]. Recent efforts have successfully applied the Lottery Ticket Hypothesis (LTH) to these models, optimizing both the model weights and masks jointly [88, 89]. A unified framework for structured pruning based on iterative training and adaptive sparsity allocation was proposed by [90]. It's worth noting that these approaches, initially designed for relatively small image-text models, encounter scalability challenges when applied to very large models.

**Hallucinations in multimodal models.** Hallucinations in multimodal models involve generating text that refers to objects not present in the input image [51, 50]. This pervasive issue affects a wide range of multimodal models, varying in architecture, training data, and scale [91, 60, 42, 6]. To gain deeper insights into this phenomenon, numerous studies have proposed evaluation benchmarks to quantify hallucinations across different dimensions [51, 50, 92, 44], shedding light on underlying causes. Specifically, co-occurrences and uncertainty [48], as well as visual uncertainty stemming from lower image resolution [93, 94, 95], have been identified as contributing factors. Additionally, it has been demonstrated that multimodal in-context learning exacerbates hallucinations [42]. To address this issue, various techniques have been proposed, including training on improved datasets [96, 44], aligning models with reinforcement learning [46, 47], refining training objectives [94], and employing post-training heuristics [48, 93]. Our study highlights the misalignment between internal representations of textual and perceptual tokens as a key cause of hallucinations.

## B  Discussion

**Study across LLMs and setups.** Our investigation primarily centers on Vicuna-v1.5 across single-task and multitask setups. We find that our conclusions remain consistent across various LLMs (e.g., OPT, Llama 2) and different settings (e.g., with and without pretraining, using different mapping modules), as detailed in the appendix. Extending our analysis to encompass other multimodal models, potentially with diverse architectures [6, 55], could offer additional valuable insights.

**Study on larger models.** While our work primarily focuses on frozen LLMs to provide insights relevant to future multimodal models, we also present results involving trained LLMs such as in LLaVA-1.5. These experiments yield observations akin to those with frozen LLM variants. However, the applicability of our experiments or the generalizability of our findings to larger LLMs (beyond 7B parameters), larger multimodal models [7, 6], or massively-trained multimodal foundation models like Gemini [97] or GPT4-V [4] remains an open question.

**Remaining questions to understand LMMs.** While we primarily investigate why and how LLMs generalize to multimodal inputs, and offer insights into issues such as object hallucinations, numerous unanswered questions persist. For instance, further exploration is needed to discern the encoded information in tokens and how LLMs extract information from visual tokens. Deeper inquiries are also required to address safety-related issues in large models, including the inability to abstain from answering, compositionality, and the precise adherence to user instructions [42].

**Other implications.** Our paper discusses several practical implications with potential benefits. Future extensions of our study could focus on specific aspects, such as enhancing model efficiency during training and inference by reducing redundant computations or model size. Additionally, addressing alignment with human preferences, such as faithfulness and safety, remains a significant challenge requiring further investigation. Our study may also inform model architecture design, such as developing mapping modules explicitly aligning multimodal tokens before entering the LLM.

## C  Broader impacts

The paper aims to enhance our comprehension of LLMs within the realm of multimodal inputs. We contend that a deeper understanding of these models can yield positive societal impacts, which we partially address in this study. For instance, our findings may contribute to mitigating the consumption of large models and their potential societal harms. Moreover, our work may inspire future research endeavors with various impacts, none of which we think must be specifically discussed here.

## D  Implementation details.

### D.1  Perceptually augmented LLM baselines

#### D.1.1  ST setup

We train many models across different datasets that span image, video and audio-text modalities. We first devise powerful baselines based on 3 tenets: (a) having the smallest number of trainable

parameters, (b) general architecture that span or similar to many existing models and (3) good performance. To this end, and inspired by previous studies showing the effectiveness of using transformer-based mapping module [10, 12, 6, 54, 55, 98], we use light-weight transformer with learnable queries and self-attention to attend to perceptual tokens. This transformer operates in low dimension space (*i.e.*, due to down/up projection layers and the the number of learnable query are limited to 10. We also favor a deeper architecture (5 blocks) compared to a wider one [10]. Our baselines are close to [10], but with significantly less trainable parameters. We train these baselines with different LLMs: OPT-6.7B [2], Llama 2-7B [5] and Vicuna-v1.5-7B [99] and different encoders for: image (ViT [23, 100], CLIP[33], MAE[101]), video (TimesFormer[24], X-CLIP[102], VideoMAE[103]) and audio (AST[25], AudioMAE[104]).

To train these baselines, we use AdamW optimizer with a learning rate of 2e-4 that decreases with a cosine annealing scheduler to a minimum of 1e-5. We train with a total batch size of 16 for captioning and 64 for VQA datasets. The number of epochs is set to 20 to ensure that all models converged, though most of these models converge after only couple of epochs. We select the best checkpoint for evaluation. For example the model for image captioning converged after $\sim$ 4 epochs. All models are trained on 8 V100 GPUs and the training time depends on the task, *e.g.*, for the large VQAv2 dataset each epoch takes $\sim$ 30 mins, for other smaller datasets it takes less time, *e.g.*, $\sim$ 10 mins for Audiocaps and MSVD-QA. Unless specified otherwise, we fix the hyperparameters for all baselines to isolate the variations that could results from this.

We refer to all text-aligned models as CLIP, trained for classification as ViT, and self-supervised with MAE objective as MAE.

### D.1.2   MT setup

To study the impact of different factors (*e.g.* pretraining, mapping module) on the internal representations (*e.g.* implicit alignment), we devise different variants of the LLaVA-1.5 [19] model that differ from the original model as follows: LLaVA-1.5-2 (LLM kept frozen), LLaVA-1.5-3 (LLM kept frozen, without pretraining) and LLaVA-1.5-4 (LLM kept frozen, without pretraining and with transformer mapping module similar. The latter is very similar to the models used in the ST setup, which ensure comparable observatoins. All these models are based on the Vicuna-v1.5-7B LLM.

We follow the same training setup of LLaVA-1.5 [19], including the training data, steps and hyperparameters.

For analysis in the paper (*e.g.* Sec. 3), we focus on Vicuna-v1.5 as it is shared by both setups. For the ST setup, we use unimodal encoders, such as ViT, TimeSformer and AST that are not aligned with text.

### D.2   Datasets and metrics

**ST setup.**   We consider a wide range of public multimodal datasets that cover 2 representative tasks: captioning and question-answering (QA) across image (VQAv2 [105], GQA [106], OKVQA [107], COCO caption [108]), video (MSVD, MSRVTQA [109], MSRVTT [110]), audio (Audiocaps [111], Clotho [112], Clotho-AQA [113]) and language tasks. For QA datasets we report the accuracy (in open-ended generation setup with exact match), and for captioning we report the CIDEr metric.

**MT Setup.**   We also evaluate the MT setup on recent datasets such as SEED [114], TextVQA [115] and POPE [50].

## E   LLMs generalize to multimodal inputs: additional experiments

### E.1   Tokens evolution across layers

**Different LLaVA-1.5 variants.**   In Fig. 13, we illustrate the differences between perceptual and textual tokens across different LLaVA-1.5 variants. Comparing the two variants with multimodal pretraining, we observe higher cross-modal similarity in LLaVA-1.5 compared to LLaVA-1.5-2, which freezes the LLM. This suggests that training the LLM enhances alignment between representations. For models without pretraining, we note that using an MLP (LLaVA-1.5-3) to connect both models yields better results than using transformer-based pooling (LLaVA-1.5-4), potentially explaining the

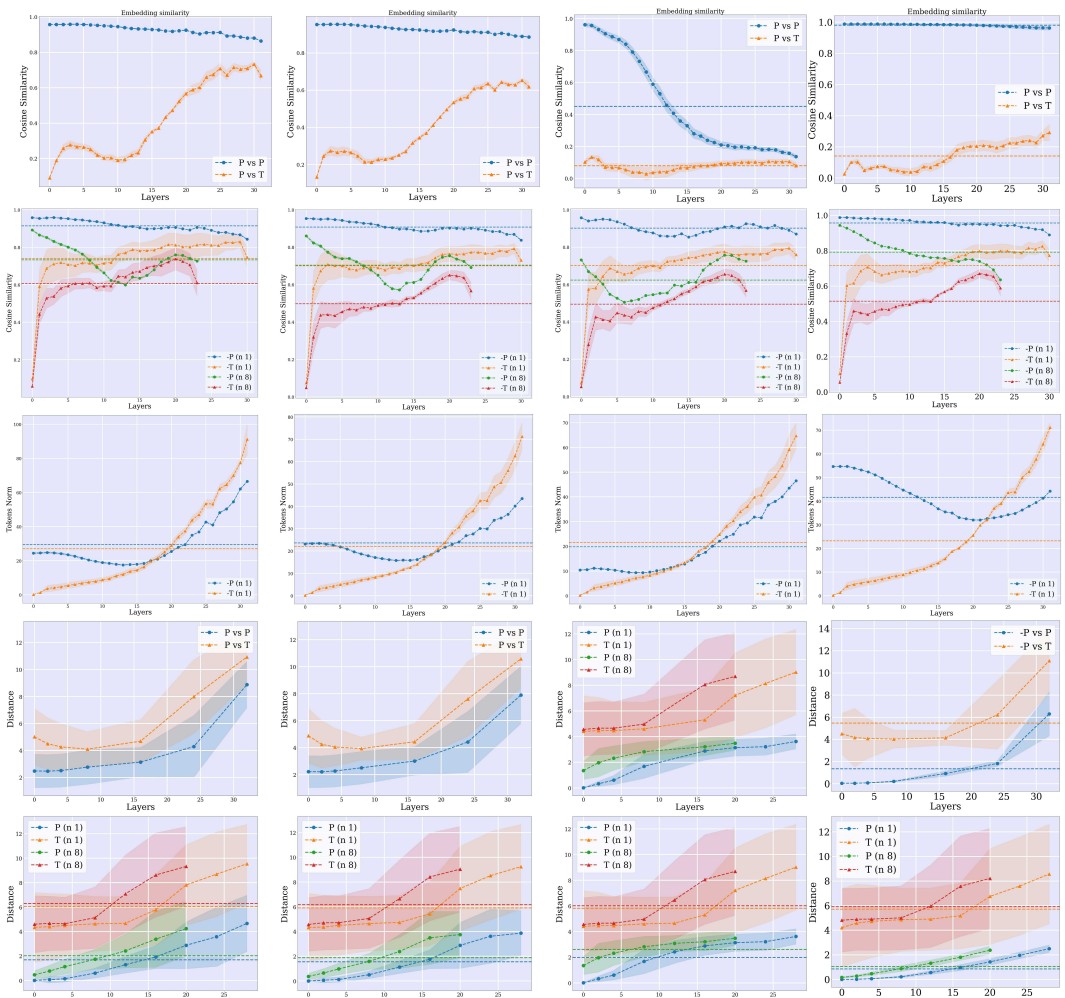

Figure 13: **Textual and multimodal tokens for LLaVA-1.5 variants (MT setup)**. From top to bottom: (1) the cosine similarity between the textual and multimodal tokens across LLM blocks. (2) the cosine similarity between consecutive blocks. (3) token norms, (4) KL-distance between vocabulary distributions decoded from textual and perceptual tokens, (6) cosine similarity between vocabulary distribution at consecutive layers. From left to right: LLaVA-1.5, LLaVA-1.5-2, LLaVA-1.5-3, LLaVA-1.5-4.

superior scores. Utilizing all visual tokens appears to bolster alignment with textual tokens, with pretraining further enhancing this alignment. Notably, vocabulary distributions undergo significant changes in middle layers, particularly for textual tokens. Similar observations hold across different variants, indicating that our findings generalize to broader setups and that training the LLM does not substantially alter token behavior.

**Different similarity measures for cross-modal alignment.** In this section, we compare the following similarity measures to compute the similarity between perceptual ($P = [p_1, ..., p_{N_p}]$) and

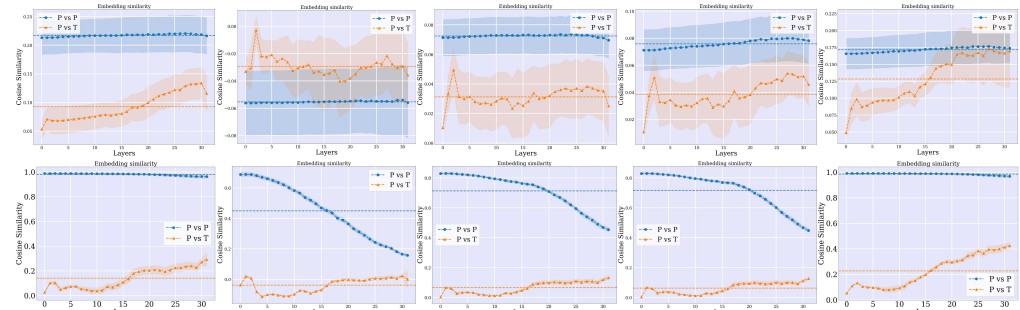

Figure 14: **Different similarity measures.** From left to rights: SimAvg, MinSim, AvgSim, MedSim, MaxSim. ST setup (top) and MT setup (bottom).

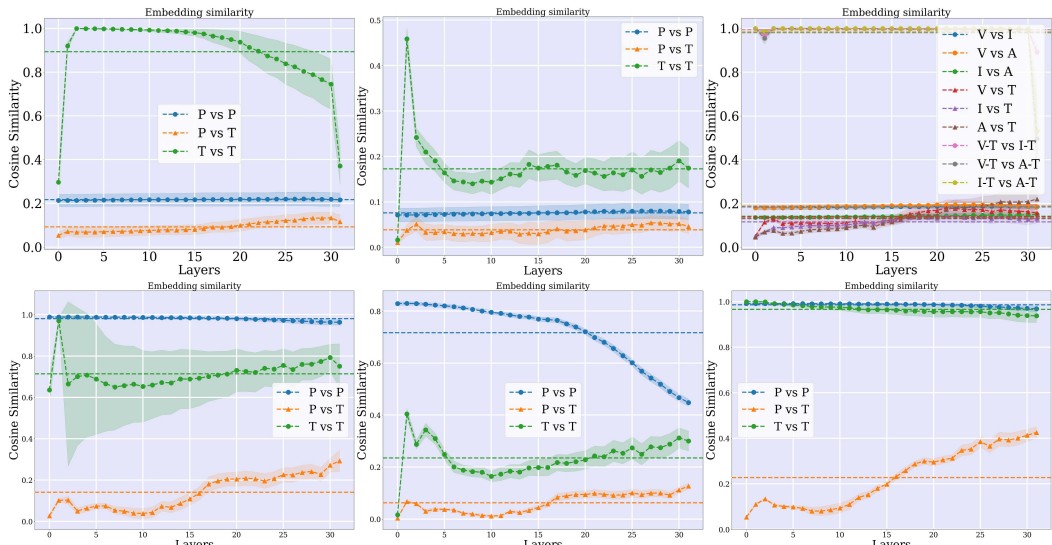

Figure 15: **Different similarity measures and the narrow cone effect.** From left to right: SimAvg, MedSim and MaxSim. Vicuna-v1.5 (top), LLaVA-1.5-4 (bottom).

textual ($T = [t_1, ..., t_{N_t}]$) tokens:

$$\text{Sim}(X, Y) = \frac{X \cdot Y}{\|X\|\|Y\|}, \tag{4}$$

$$\text{SimAvg}(P, T) = \text{Sim}(\hat{P}, \hat{T}), \quad \hat{P} = \frac{\sum_i^{N_p} p_i}{N_p}, \quad \hat{T} = \frac{\sum_i^{N_t} t_i}{N_t}, \tag{5}$$

$$\text{MaxSim}(P, T) = \max_{i \in [N_p] j \in [N_t]} \text{Sim}(p_i, t_j), \tag{6}$$

$$\text{MinSim}(P, T) = \min_{i \in [N_p] j \in [N_t]} \text{Sim}(p_i, t_j), \tag{7}$$

$$\text{AvgSim}(P, T) = \frac{\sum_{i \in [N_p] j \in [N_t]} \text{Sim}(p_i, t_j)}{N_p + N_t}, \tag{8}$$

$$\text{MedSim}(P, T) = \text{Med}_{i \in [N_p] j \in [N_t]} \text{Sim}(p_i, t_j), \tag{9}$$

Where $[N_p] = \{1, ..., N_p\}$ and $[N_t] = \{1, ..., N_t\}$ and Med is the median operation.

Fig. 14 shows the inter ($P$ vs $T$) and intra ($P$ vs $P$) similarity. According to all measures, except AvgSim and MinSim, we have similar observations: increasing inter similarity and higher intra similarity that increases in last layers. For MinSim and AvgSim for the ST setup, we do not see such observations, indicating that not all perceptual tokens are, or should be aligned to text.

Fig. 15 and Fig. 16 show the measures comparison for the narrow cone experiments. Interestingly, the narrow cone effect is less seen when looking at the median of the token similarities (MedSim), indicating that this effect is not driven by all tokens, and at the token level the representation is not always anisotropic.

In spite of having similar observations between several measures, we focus on SimAvg, as it is much faster to compute, especially whey there is large number of tokens (as in LLaVA-1.5).

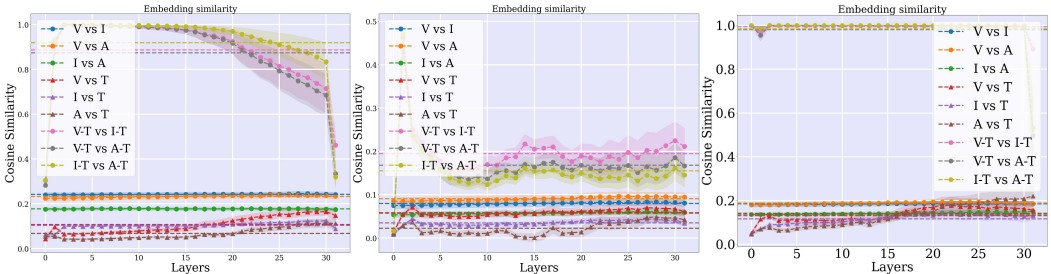

Figure 16: **Narrow cones for image, video, audio and text modalities.** From left to right: SimAvg, MedSim and MaxSim. Vicuna-v1.5 (top), LLaVA-1.5-4 (bottom).

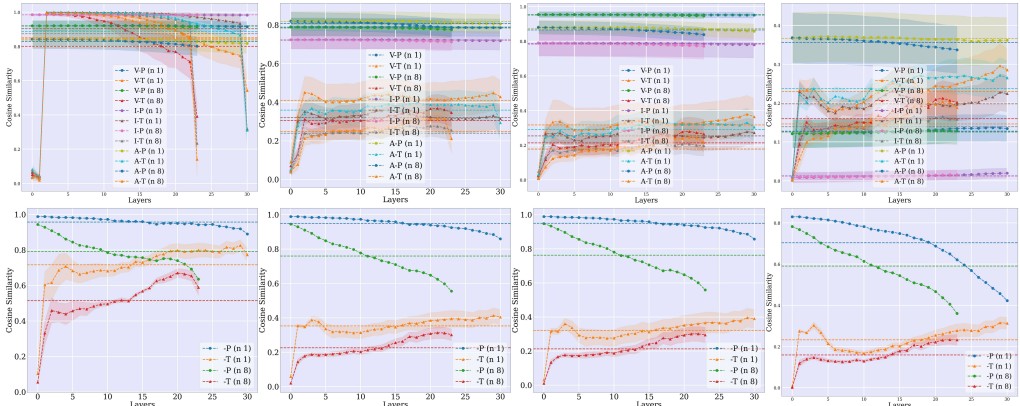

Figure 17: **Different similarity measures between tokens at consecutive layers.** From left to right: SimAvg, AvgDiagSim, MedDiagSim and MedSim. Vicuna-v1.5 (top), LLaVA-1.5-4 (bottom).

**Different similarity measures for the similarity across consecutive layers.** we compare the following similarity measures to compute the token similarity between consecutive blocks (*e.g.* between tokens at block a $X^a = x_1^a, ..., x_N^a$ and b $X^b = x_1^b, ..., x_N^b$):

$$\text{Sim}(X^a, X^b) = \frac{X^a \cdot X^b}{\|X^a\|\|X^b\|}, \tag{10}$$

$$\text{SimAvg}(X^a, X^b) = \text{Sim}(\hat{X}^a, \hat{X}^b), \quad \hat{X} = \frac{\sum_i^N x_i}{N}, \tag{11}$$

$$\text{AvgDiagSim}(X^a, X^b) = \frac{\sum_{i \in [N]} \text{Sim}(p_i^a, p_i^b)}{N}, \tag{12}$$

$$\text{MedDiagSim}(X^a, X^b) = \text{Med}_{i \in [N]} \text{Sim}(p_i^a, p_i^b), \text{MedSim}(X^a, X^b) = \text{Med}_{i,j \in [N]} \text{Sim}(p_i^a, p_j^b), \tag{13}$$

Where $[N] = \{1, ..., N\}$ and Med is the median operation.

Fig. 17 shows similar observations across all different measures when each token is compared with the token at the same position in different layers. However, when taking the median of similarities (MedSim) across all tokens, this similarity is siginfcantly smaller, especially for the ST setup. This reveals that tokens can be very different within the same modality or example.

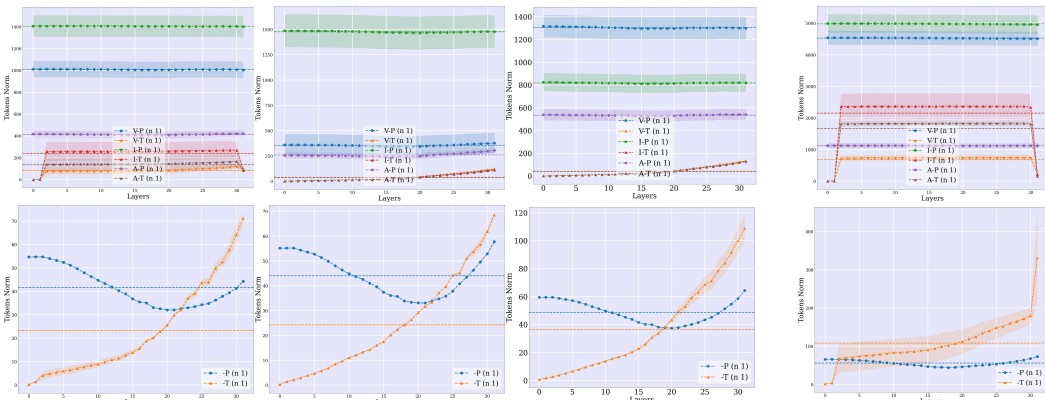

Figure 18: **Different token norm measures**. We compute the token L2 norm at consecutive blocks (e.g. $B^{l+n}$ and $B^l$) for the ST (top) and MT (bottom) setups. From left to right: NormAvg, MinNorm, MedianNorm and MaxNorm.

**Tokens evolution for different modalities.** Fig. 19, shows that textual and multimodal tokens evolve differently inside LLMs.

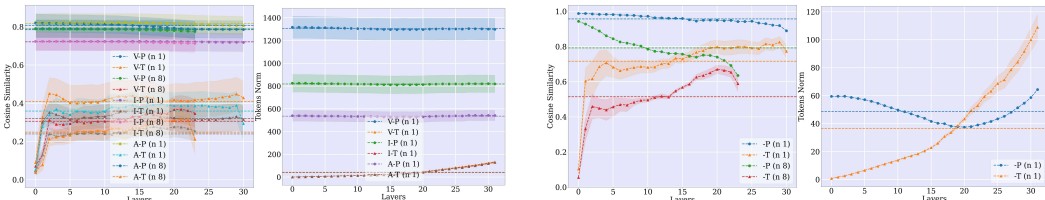

Figure 19: **Textual and multimodal tokens evolve differently inside LLMs**. We compute the tokenwise cosine similarity and the median token L2 norm at consecutive blocks (e.g. $X^{l+n}$ and $X^l$) for the ST (left) and MT (right) setups.

## E.2 Token norms across layers

**Massive token norms.** In this section we highlight the presence of tokens with massive norms, this becomes clearer when looking at different norm measures. We compare the following measures to compute the token L2 norms across blocks (*e.g.* $X = x_1, ..., x_N$):

$$\text{Norm}(X) = \sqrt{\sum_i^M X_i^2}, \tag{14}$$

$$\text{NormAvg}(X) = \text{Norm}(\hat{X}), \quad \hat{X} = \frac{\sum_i^N x_i}{N}, \tag{15}$$

$$\text{MinNorm}(X) = \min_{i\in[N]} \text{Norm}(x_i), \tag{16}$$

$$\text{MedianNorm}(X) = \text{Med}_{i\in[N]}\text{Norm}(x_i), \tag{17}$$

$$\text{MaxNorm}(X) = \max_{i\in[N]} \text{Norm}(x_i), \tag{18}$$

Where $[N] = \{1, ..., N\}$ and Med is the median operation and M is the total number of elements in the tensor. For the ST setup, Fig. 18 shows a very high token norms when looking at NormAvg and MaxNorm, compared to MinNorm and MedianNorm. These massive norms are present for both textual and perceptual tokens, and they are larger for perceptual ones. When looking closely, we find that these tokens correspond to start or split tokens as seen in [34]. For the MT setup, we notice that these massive tokens presents mainly in the system message, which we remove for our study as it is common for all examples. Interestingly the perceptual tokens for the MT setup do not seem to have massive norms.



Figure 20: **L2 token norm increases with training**. We compute the token L2 norm during training and across the LLM blocks for the ST setup (Vicuna-v1.5). From left to right: NormAvg, MinNorm, MedianNorm and MaxNorm.

**Increasing token norm during training.** We try to investigate why we have very high perceptual token norms. To this end, we compute the norm across different epochs. Fig. 20 shows that during training of the mapping module, the norms increase significantly.

### E.3 Vocabulary distribution

For each token, we use the LLM unembedding (*i.e.* LLM head) to decode the latent representation to a probability distribution over the vocabulary. This approach have shown to work well for LLMs at different layers, not just the last one [35, 36, 37, 38]. In Fig. 21, we show the histogram of this distribution at the first LLM layer for both textual and perceptual tokens, the KL-distance between the 2 distributions, KL-distance between consecutive layers and the entropy. Here we report additional results for the LLaVA-1.5 baseline showing similar observations to those of ST reported in the main paper.

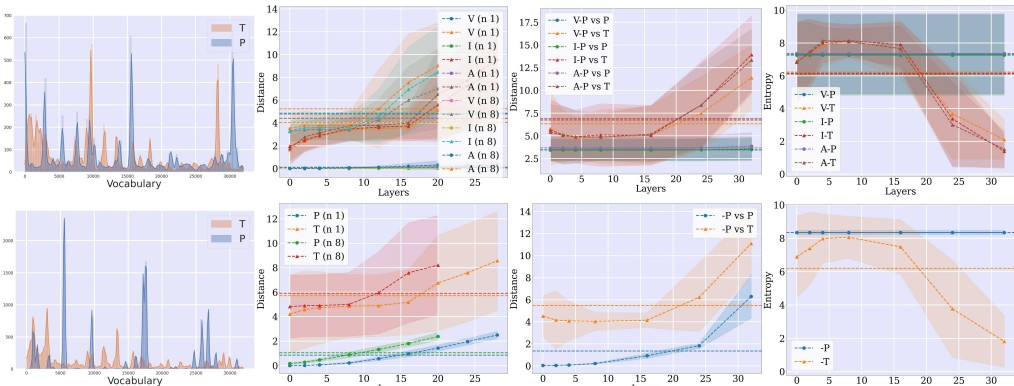

Figure 21: **Textual and visual tokens have different vocabulary distributions inside LLMs.** We use the LLM unembedding layer to map each token to a probability distribution over the vocabulary. We then show (from left to right): the histograms at the input of the LLM, the KL divergence between the distributions at consecutive layers, the KL divergence between textual and perceptual distributions and the distribution entropy. Top: Vicuna-v1.5, Bottom: LLaVA-1.5-4.

### E.4 Similar activated weights by different modalities

**Experimental setup.** In this section, we analyse the subnetworks activated by different multimodal inputs. We use the Wanda score [39] to extracted these subnetworks or pruning masks, then compute the IoU. For multimodal datasets we consider only the perceptual tokens, for example the visual tokens without the questions for VQAv2. We also use the text in these datasets as a source for textual tokens (*e.g.*, COCO-text consider only the captions in the COCO dataset).

**Different LLaVA-1.5 variants.** In Fig. 23, we show the overlap between the weights activated by different modalities. Across different LLaVA-1.5 variants, we find similar observations: high overlap between perceptual and textual activated weights (*e.g.* 0.6 IoU), which is less than the overlap between weights acrtivated by the same modality (*e.g.* 0.95 for perceptual tokens and 0.87 for textual

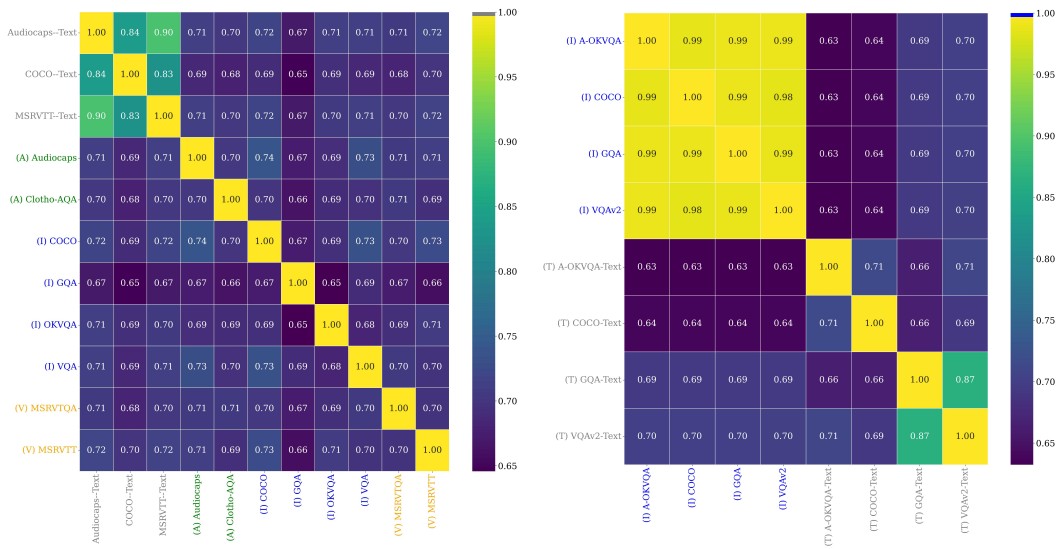

Figure 22: **IoUs of multimodal subnetworks**. IoU of the subnetworks activated by different tasks and modalities, for the ST (left) and MT (right) setups. Different modalities activate similar LLM weights.

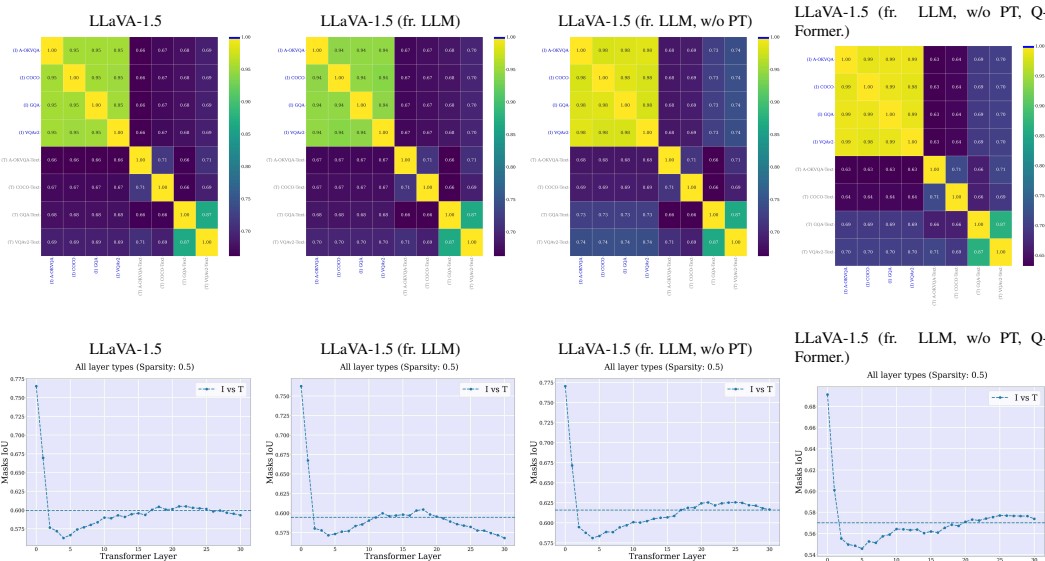

Figure 23: **IoUs of activated subnetworks for LLaVA-1.5 variants**. We compute the IoU of weights activated by different multimodal tokens. From left to right: LLaVA-1.5, LLaVA-1.5-2, LLaVA-1.5-3, LLaVA-1.5-4.

ones). We also notice a significant decrease in IoU at the first layers, which might reveal that the first layers encode general features that are shared across modalities.

**Different LLMs for the ST setup.**    In Fig. 24, we show the IoUs of activated weights for different frozen LLMs (*i.e.*, OPT, Llama 2 and Vicuna-v1.5) for the ST setup. We notice similar observations across LLMs, and relatively higher overlap for OPT. We notice similar observation compared to the MT setup, where we have a significant decrease in the IoU in first layers. For this setup, it is clearer that the overlap increase for deeper layers.

**Different sparsity levels.**    In Fig. 25, we study how the overlap between activated weights changes with the size of the extracted subnetworks. This size depends on the sparsity of the final model. We

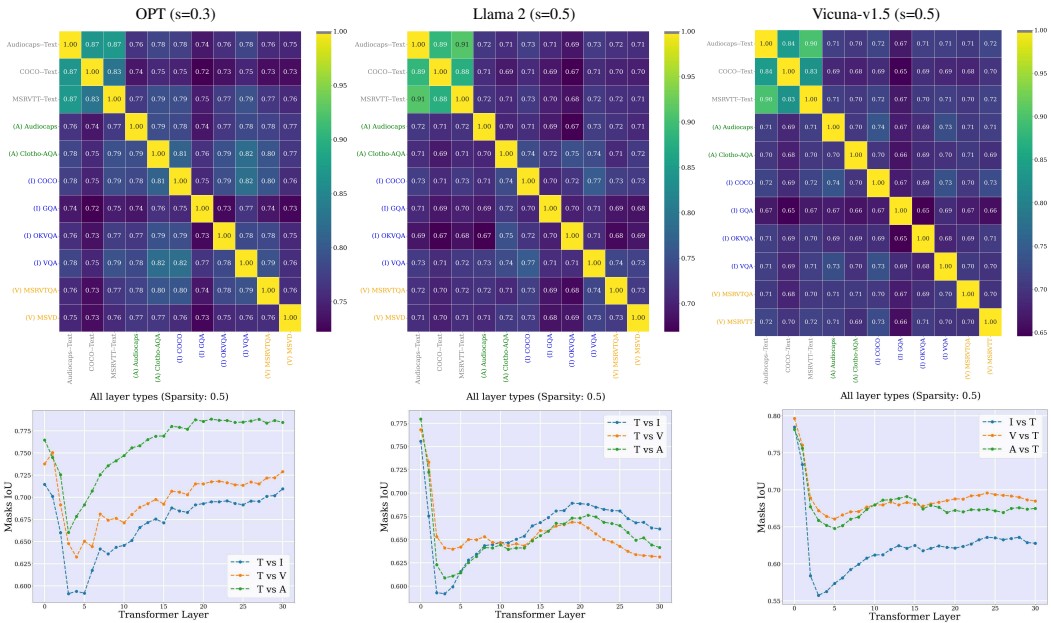

Figure 24: **IoUs of activated subnetworks for different LLMs**. We compute the IoUs for weights activated by different multimodal tokens. From left to right for the ST setup: OPT, Llama 2, Vicuna-v1.5.

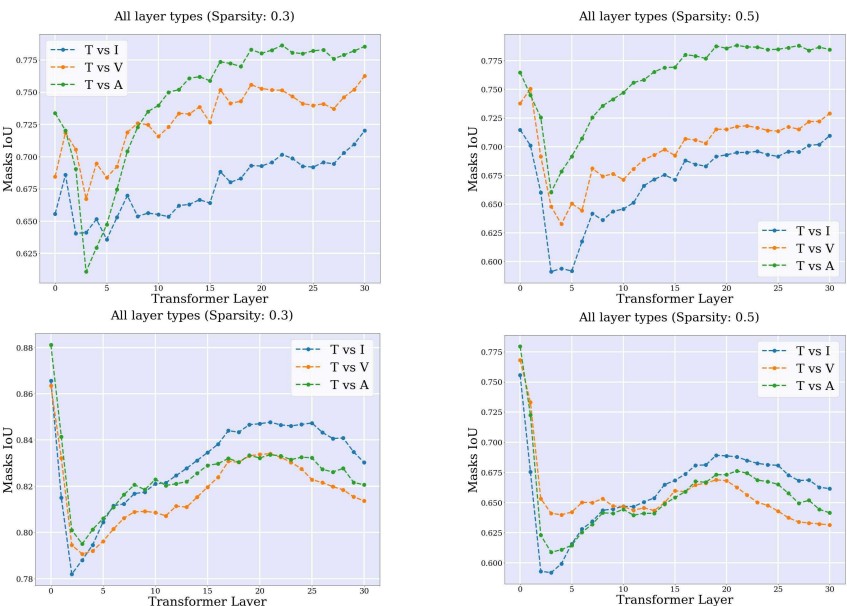

Figure 25: **Overlap between multimodal subnetworks at different sparsity levels.**. We compute the IoU of activated weights across layers at 0.3 and 0.5 sparsity levels. OPT (top), Llama 2 (bottom).

notice that the lower the sparsity, the higher the overlap, revealing that higher sparsity allows to extract more modality-specific activated weights.

**Pruning weights by streaming different modalities.** In addition to the IoU, we also compare the differences between the task performance of activated weights. Fro each dataset, we give to the model either the perceptual prompt, the textual prompt or both, knowing that each example in the dataset consists of a perceptual prompt followed by the textual one. Fig. 26 shows slight differences in overall performance when the LLM is pruned by different modalities, with the best performance is

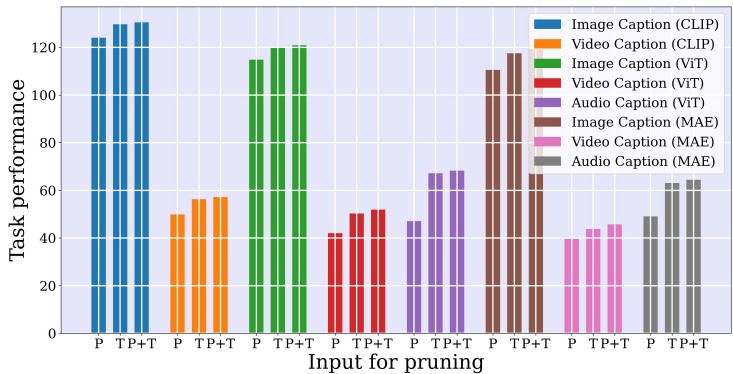

Figure 26: **Similar LLMs weights are activated by different modalities**. We report the task performance when keeping only the subnetwork activated by: multimodal prompt (P), text (T). or both (P+T) for OPT. Sparsity level: 0.3.

by considering both the textual and the perceptual tokens. This also show the high overlap between the weights used to activate textual and perceptual tokens.

**Transfer of pruning masks to other tasks and modalities.** To further highlight the overlap between weights, we report the performance when the model is pruned given data from other modalities or datasets. We notice similar observations across LLMs, such Llama 2 Fig. 29 and Vicuna-v1.5 Fig. 30. Interestingly, we find the similar overlap with the unsupervised MAE encoders Fig. 27 compared to text aligned ones Fig. 28. We notice a performance degradation when the model is pruned at high sparsity levels (0.5).

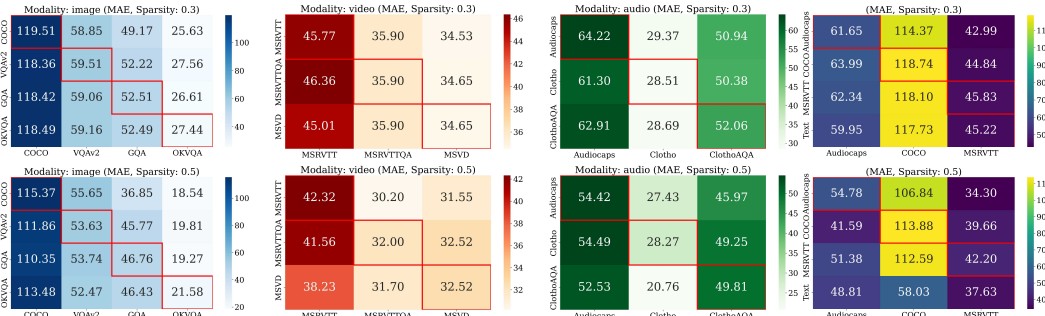

Figure 27: **Transfer of multimodal subnetworks across tasks and modalities with OPT and MAE encoders**. We use the subnetwork activated by a given task/modality to other tasks/modalities and report the task performance. From left to right, transfer across: image tasks, video tasks, audio tasks and across modalities for the captioning task. In each figure, the row corresponds to the source dataset of the subnetwork and the column to the target dataset.

**Modality-specific subnetworks?** The experiments suggest a high overlap between weights activated by different modalities. For instance, the pruning masks similarity (IoU) between datasets within the same modality is on par with those across modalities for the ST setup. However, this does not exclude the possibility of finding weights that are generally activated when seeing a particular modality, even if there are small amount of them. The overlap is smaller with LLaVA-1.5 variants making this possibility more likely for large scale multitask models.

## E.5 Implicit multimodal alignment effect

**Alignment inside each LLM Block.** Fig. 32 reports the tokens similarity and norms for both LLaVA-1.5 and Vicuna-v1.5.

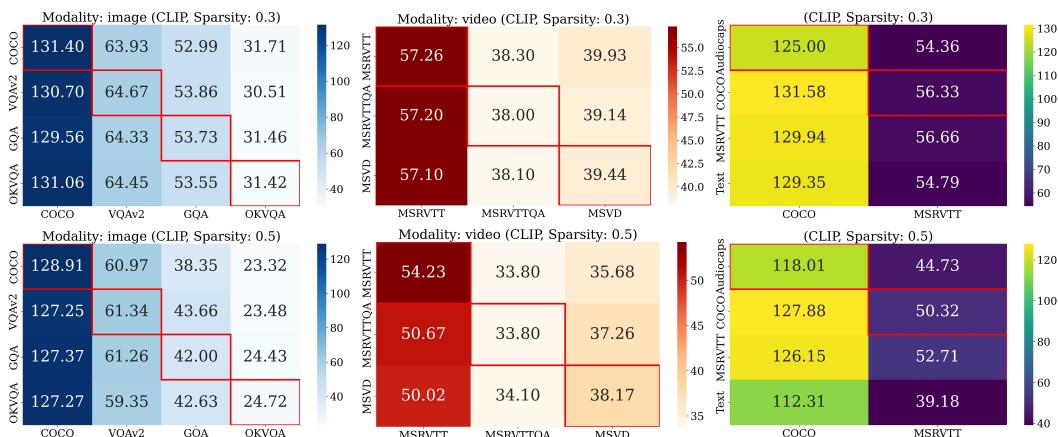

Figure 28: **Transfer of multimodal subnetworks across tasks and modalities with OPT and CLIP encoders**. We use the subnetwork activated by a given task/modality to other tasks/modalities and report the task performance. From left to right, transfer across: image tasks, video tasks, audio tasks and across modalities for the captioning task. In each figure, the row corresponds to the source dataset of the subnetwork and the column to the target dataset.

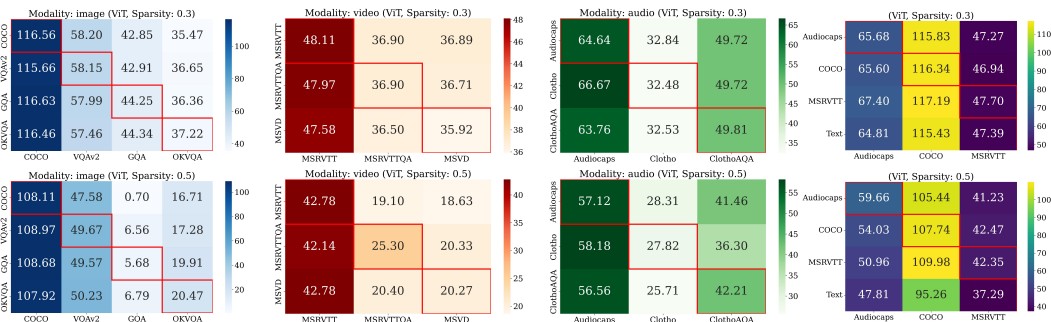

Figure 29: **Transfer of multimodal subnetworks across tasks and modalities with Llama 2 and ViT encoders**. We use the subnetwork activated by a given task/modality to other tasks/modalities and report the task performance. From left to right, transfer across: image tasks, video tasks, audio tasks and across modalities for the captioning task. In each figure, the row corresponds to the source dataset of the subnetwork and the column to the target dataset.

## F   Implications on performance, safety and efficiency: additional experiments

### F.1   Implicit multimodal alignment as proxy metric for task performance?

**IMA score across epochs.**   Fig. 33 shows an increasing similarity between textual and perceptual tokens during training with OPT and Vicuna-v1.5.

Table 1: **IMA score across different encoders.** We report the IMA score and the task performance with the ST setup (OPT). A positive correlation exists between IMA score and the performance; the most aligned encdoers (CLIP) have the best accuracy/CIDEr on VQA and captioning tasks.

| LLM | Encoder | IMA Score ↑ | COCO ↑ | VQAv2 ↑ | GQA ↑ |
|---|---|---|---|---|---|
| | | | CIDEr (test) | Acc (Val) | Acc (Val) |
| Vicuna-v1.5 | CLIP-ViT-L | 0.130 | 127.63 | 63.05 | 54.34 |
| Vicuna-v1.5 | ViT-L (ImageNet) | 0.105 | 116.76 | 61.27 | 51.57 |
| Vicuna-v1.5 | MAE-L | 0.060 | 76.40 | 59.57 | 52.88 |

**IMA score across different encoders.**   In Table 1 we compare different image encoders and report the the IMA score and the task performance on several VQA and image captioning tasks. Encoders

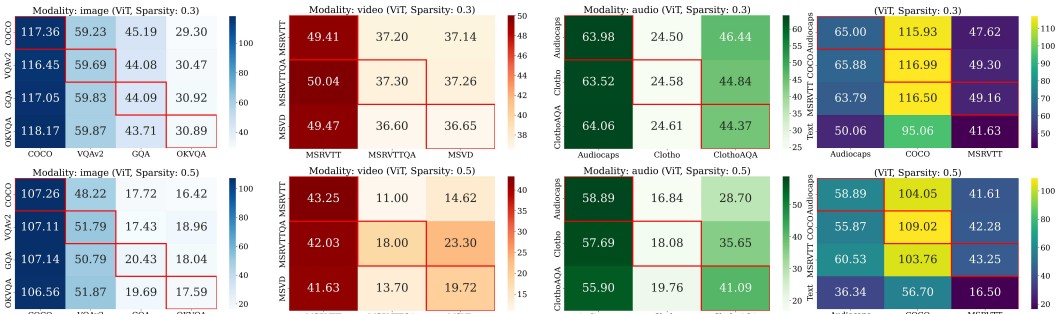

Figure 30: **Transfer of multimodal subnetworks across tasks and modalities with Vicuna-v1.5 and ViT encoders**. We use the subnetwork activated by a given task/modality to other tasks/modalities and report the task performance. From left to right, transfer across: image tasks, video tasks, audio tasks and across modalities for the captioning task. In each figure, the row corresponds to the source dataset of the subnetwork and the column to the target dataset.

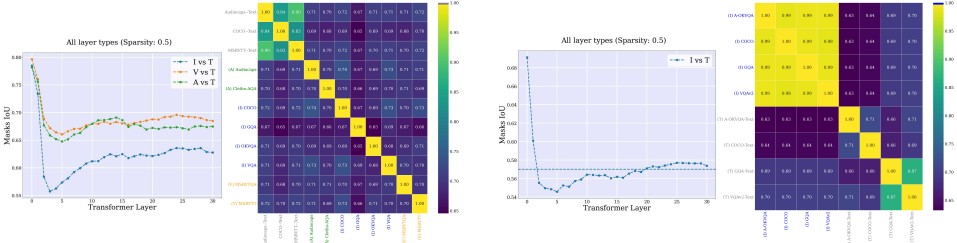

Figure 31: **High similarity between LLM weights activated by different modalities**. We compute the IoU of the subnetworks activated by different tasks across modalities, for the ST (left) and MT (right) setups.

that are most aligned to textual tokens inside LLMs (highest IMA *e.g.* CLIP) have also the best task performance.

### F.2    Implicit multimodal alignment as proxy metric for hallucination?

We provide more details reagarding the hallucinations metrics. These metrics are supposed to measure multimodal or object hallucination.

**CHAIR on COCO captioning [51].**    On the COCO image captioning dataset, the model is asked to describe the images. We compute the CHAIR metrics based on the generated captions and the ground truth annotations of all objects in the image. If a caption contains non-existent objects, we classify it as a hallucinated caption. The CHAIRs score is the ratio of hallucinated captions to the total number of captions. Additionally, we calculate the ratio of hallucinated objects to the total number of objects across all captions, which is referred to as CHAIRi. A CHAIR score of 0 indicates no hallucinations. In the paper, we report (1 - CHAIR) × 100, thus a higher score indicates fewer hallucinations.

**POPE benchmark [50].**    This is a question-answering task involving questions about the existence of objects in images. The metric used is accuracy; the fewer the hallucinations, the higher the accuracy.

### F.3    Skipping computations for visual tokens.

In this section, we propose to skip computations for the visual tokens.

**Skip FFN Tokens.**    We randomly skip a number of tokens (we refer to this amount as skip ratio), both the textual and the remaining visual tokens are processed in the FFN layers. **??**, shows a linear relationship between the skip ratio and task performance, the higher the ratio the lower the scores.

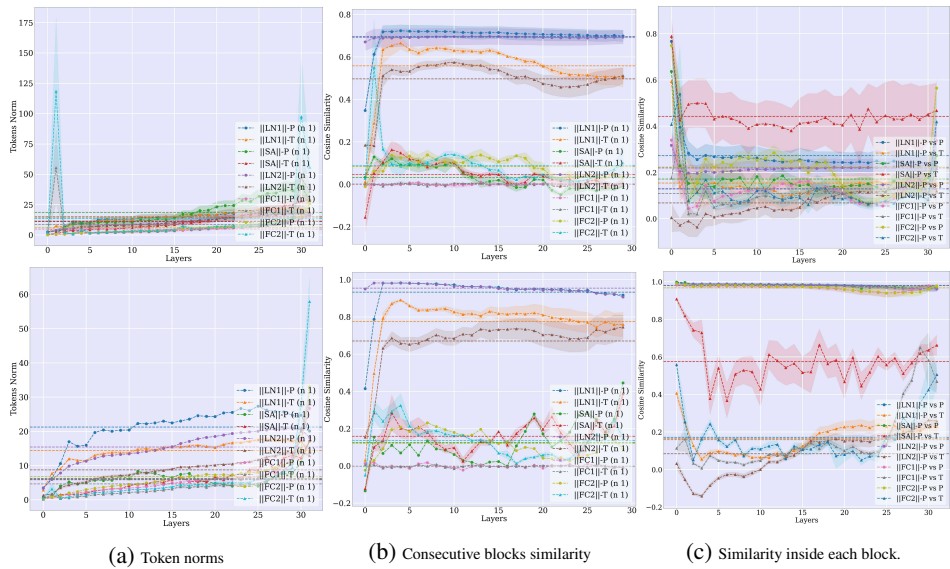

| (a) Token norms | (b) Consecutive blocks similarity | (c) Similarity inside each block. |

Figure 32: **Implicit alignment inside the LLM blocks.** We compute the token norms (left), tokens cosine similarity between consecutive blocks (middle) and across modalities (last). The tokens are inside the LLM blocks (and outside the residual stream): after the self-attention (SA), and FFNs (FC1/2) and layer norms (LN). From top to down: Vicuna-v1.5, LLaVA-1.5-4.

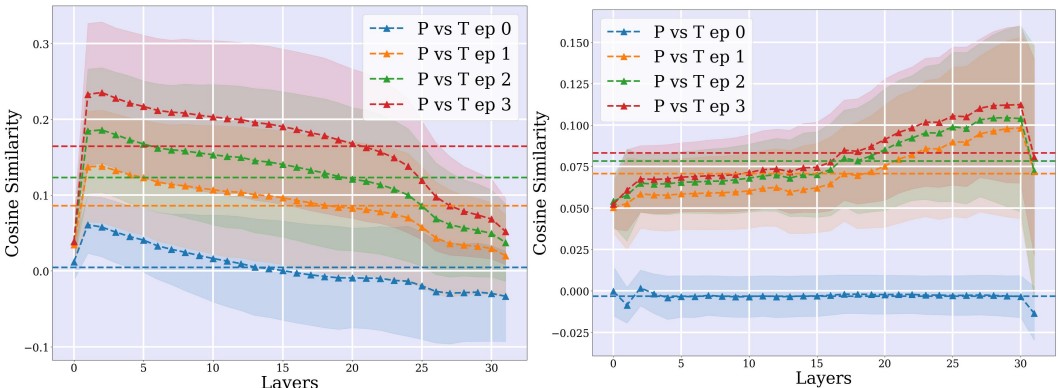

Figure 33: **Implicit alignment score across epochs.** We report the implicit alignment score for OPT (left) and Vicuna-v1.5 right) during training of the mapping module.

## F.4 α-SubNet

In Table 2, we evaluate our α-SubNet on additional multimodal tasks. Compared to other task-agnostic baselines such as magnitude pruning the scores of α-SubNet are significantly higher. This support more that the multimodal tokens activate similar weights inside LLMs.

Table 2: **α-SubNet a task and modality-agnostic subnetwork.** We prune the LLMs using different post-training pruning methods, including our α-SubNet.

| Method | #P/#TP/Sparsity | Avg | COCO ↑ | VQAv2 ↑ | OKVQA ↑ | GQA ↑ | MSR-VTT ↑ | MSRVTT-QA ↑ | MSVD-QA ↑ | Audiocaps ↑ | Clotho ↑ | Clotho-AQA ↑ |
| | | | CIDEr (test) | Acc (Val) | Acc (Val) | Acc (Val) | CIDEr (test) | Acc (test) | Acc (test) | CIDEr (test) | CIDEr (test) | Acc (test) |
|---|---|---|---|---|---|---|---|---|---|---|---|---|
| MAPL [12] | 7B/3.4M/0.00 | – | 125.2 | 43.5 | 18.7 / 31.6 | – | – | – | – | – | – | – |
| eP-ALM [9] | 6.7B/4M/0.00 | – | 111.6 | 54.9 | – | 42.91 | 48.79 | 35.90 | 38.40 | 61.86 | – | – |
| DePALM [10] | 7B/18.1M/0.00 | – | 131.29 | 70.11 | 37.69 | – | 49.88 | – | – | 69.70 | – | – |
| Baseline | 6.7B/7M/0.00 | 57.71 | 132.83 | 63.49 | 33.01 | 55.29 | 58.23 | 38.84 | 38.83 | 68.24 | 35.66 | 52.72 |
| Wanda | 6.7B/7M/0.50 | 51.32 (88.93%) | 126.81 | 55.28 | 24.72 | 42.00 | 54.23 | 33.80 | 37.17 | 58.99 | 31.81 | 48.41 |
| Random mask | 6.7B/7M/0.47 | 0.00 (0%) | 0.00 | 0.00 | 0.00 | 0.00 | 0.00 | 0.00 | 0.00 | 0.05 | 0.00 | 0.00 |
| α-SubNet (s=0.3) | 6.7B/7M/0.47 | 39.34 (68.17%) | 106.77 | 51.77 | 17.72 | 38.09 | 38.37 | 29.80 | 31.19 | 23.15 | 8.52 | 48.03 |

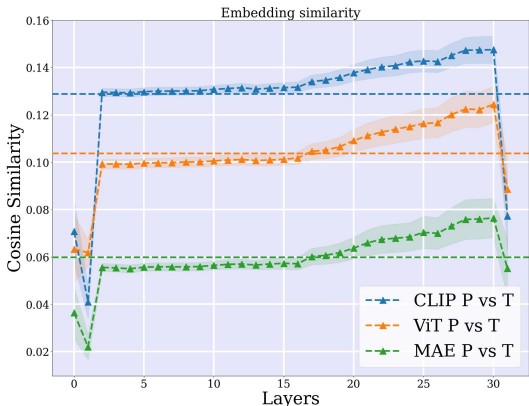

Figure 34: **Comparison of implicit multimodal alignment score across layers for different encoders.** CLIP models produce features that are most aligned to textual tokens across LLM layers. On the other hand, self-supervised encoders (*e.g.* MAE) produce the least text-aligned features. However, the relatively low cosine similarity score (closer to 0), reveals that the modality gap (*e.g.* Narrow cones) still exists in LLMs, even for text-aligned encoders.

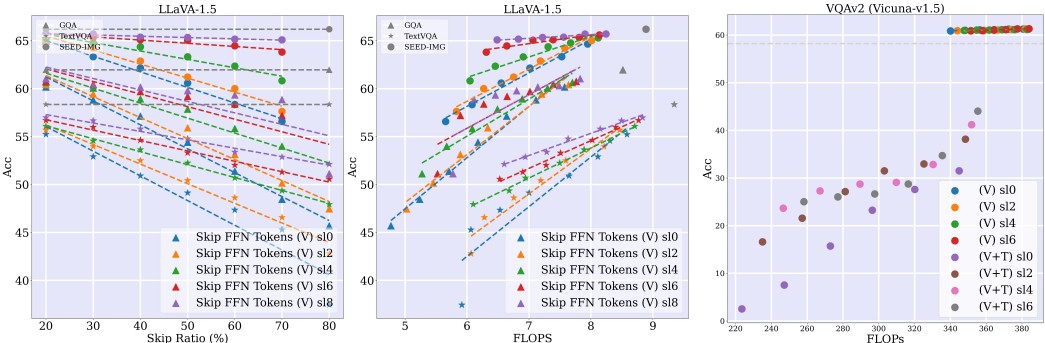

Figure 35: **Skipping computations for visual tokens**. Skip Tokens: we skip (Skip ratio)% of the tokens in the FFN layers. sl: skipping start layer. (V): visual tokens. (T): textual tokens. Results MT (with LLaVA-1.5) and ST (last column) setups.

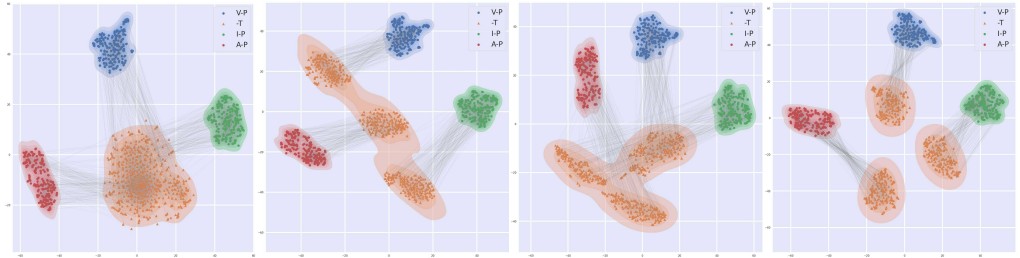

Figure 36: **t-SNE visualization of tokens inside LLMs.** From left to right: layer 0, 1, 24 and 32 for Vicuna-v1.5.

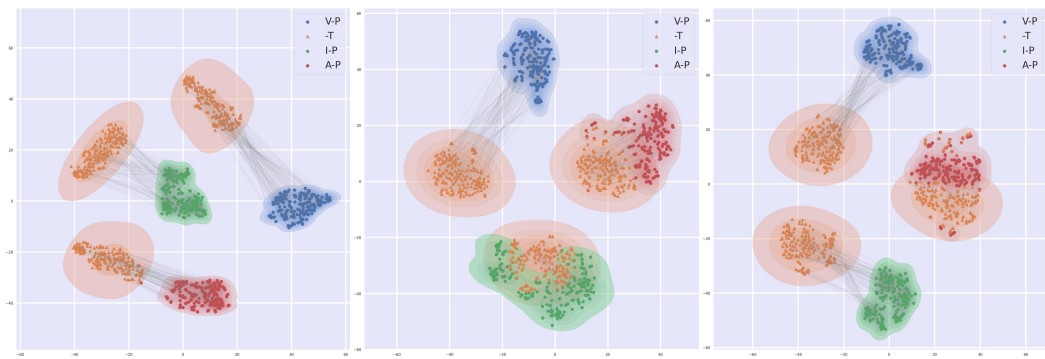

Figure 37: **t-SNE visualization of tokens inside LLM blocks (after SA layers).** From left to right: layer 0, 1, 24 and 32 for Vicuna-v1.5. There is less separation inside the LLM block.

