# OpenReview forum: "Implicit Multimodal Alignment: On the Generalization of Frozen LLMs to Multimodal Inputs"
_NeurIPS.cc/2024/Conference — NeurIPS 2024 poster_

### Official Review · Reviewer_69kn · 2024-07-04

**Soundness:** 3
**Presentation:** 1
**Contribution:** 4
**Rating:** 6
**Confidence:** 4

**Summary:**

The authors propose an exploration of the Implicit Multimodal Alignment (IMA) phenomenon in frozen Large Language Models (LLMs): when exposed to perceptual tokens (e.g. from image or audio features), authors show that those tokens are implicitly aligned on text tokens, even so the LLM has only been trained on text previously. The study argues this phenomenon is a fundamental mechanism into LLMs, and that it is relevant to (i) better understand task performance (e.g using that implicit alignment as proxy) and (ii) to better understand model visual faithfulness (i.e absence or presence of hallucinations). Based on their findings, the authors propose as well architecture changes such as skipping specific operations on perceptual tokens in order to decrease inference cost.

**Strengths:**

- This is a very original and in-depth work to understand the inside mechanisms appearing in MLLMs. Many/Most of the current published MLLMs are based on variations a pre-trained LLM + a pre-trained vision encoder with smaller VL connector in between. The insights from this study could be applied on many of those works, especially to better improve the visual faithfulness of those models.

**Weaknesses:**

- This is not commonly an important weakness but for this work, it might be: the presentation of the results and the overall quality of the layout organization is hurting the understanding. All the figures are pixelated bitmap/jpeg screenshots and are barely readable. This doesn't change the pertinence of the work, but hurt importantly how well its content is conveyed. I urge the authors to revisit this. Many solutions exist today, such as TKIZ or exporting figures in vectorized PDF to better include them in the final document. Lacking of space in a limited-page paper is understandable, but not using TKIZ or vectorized figures is not. Figure 5 for instance shouldn't be as is in this work. I challenge other readers to read/decipher the labels of the x and y axes on the heatmaps.

- In many details, the work could have be improved. See the "Typos" list below. As well, some claims are not backed anywhere. For instance, Figure 1 and Figure 12 both claim a 70% sparsity. Where is that introduced in the main text? Consistency is not respected as well: in some contexts, P means Prompt while in other context means Perceptual. Is Prompt == Perceptual? This is not stated in Section 2. Prompt usually refers to the text and image prompts, before preprocessing and tokenization. The formatting is off as well. Sometimes, $P$ is used, sometimes plain-text P. Same for $T$ vs. T.

- Evaluation could be more complete for the Multi Task setup with the use of common benchmarks such as LLaVABench-in-the-Wild, MMVet, etc. As well, Hallucinations benchmarks such as MMHALBench and HallusionBench should be used. Authors quote [46], which is the very work that introduces MMHALBench.

Typos:
 - Line 90: Define k?
 - Line 94: FC1 and FC2 should be $FC1$ and $FC2$. Same for g, LN1, LN2, SA.
 - Line 92: What is (B)? It is not defined in eq (2).
 - Line 101: s/We/we/
 - L105: Reformulate "In the paper, we focus no LLaVA-1.5-4 as it is most similar to the ST setup, and analyse other variants in App. E."? s/ no / on / ?
 - L106: (e.g. Sec. 3) ? Do you mean (see Sec. 3)? Why e.g.?
 - Section/Equation/Appendix references are not following the same format. Sec. is used in place of Sec., same for App. and Appendix, but Equation is fully written. What about adopting a consistent terminology?
- Figure 2 is barely readable, same for Figure 3, Figure 4. Figure 5, as well it is not vectorized. Consider using TKIZ or export your figures in PDF and include those cropped PDF in your tex document. Bitmap/JPEG/PNG used here are leading to poor and pixelated results, hurting the readability.
- Figure 1: s/Analysing/Analyzing/
- Figure 1: s/LMM/LLM/ (twice)
- Figure 1: s/perceputa/perceptual/
- Figure 1: "Computation efficiency", do you mean "Computational efficiency" or "Computation and efficiency"?
- Figure 1: What is the point of the right block named "Framework"? It doesn't seem related to the work or the caption of Figure 1.
- L136: The claim "Fig. 2 shows a clear narrow cone effect for textual and perceptual tokens." is not verifiable when looking at Figure 2.
- Figure 10. What is the difference between Cosine Similarity and Sim. Aren't they the same as defined in Equation 3? Why using different terminology?

**Questions:**

- Author claim that the alignment between Image <-> Text tokens helps the model performing well on VQA tasks and hallucinations tasks. It would be interesting to know if this is a cause (i.e performance on vision tasks is high thanks to that alignment), or a consequence (i.e alignment is high along performance on vision tasks thanks to the vision-language training). A way to verify that could be to compare alignment among different off-the-shelf models (e.g. LLaVA-1.5, LLaVA-1.6, Qwen-VL-Chat, Phi3-Vision) and see if there are differences in their respective alignment scores. With such population of models and scores, it could be possible to rely on statistical testing to establish it. Do you see a covariation between alignment and performance among that population? Can you control the alignment independently of the training on vision-language itself? E.g. manipulating directly P and T such as the similarity is higher/lower, and still observe that covariation between the alignment and performance variables in that population?

 - As well, the authors claim that an LLM can be seen as a residual stream with refinement blocks acting as steering blocks, and that this architecture is what helps the LLM to implicitly having an alignment between text and image tokens. It seems hard to support this claim without contrasting their experiments with another architecture aligning image and text. For instance, using a simple pre-trained CNN to process the image, and a simple pre-trained RNN (e.g. an LSTM or GRU) to process the text input, and then concatenating/pooling the outputs with a simple connector (e.g c-abstractor or just a plain linear projection) could be used to contrast their findings. Is that something planned by the authors?

**Limitations:**

- One important limitation is the presentation of this work. The majority of the figures are pixelated and not readable. This is often a minor inconvenience in other works, but in this paper all the results are presented through figures. Figure 5 for instance should be removed or importantly reworked. Lacking of space in a limited-page paper is understandable, but not using TKIZ or vectorized figures is not. This is hurting the overall understanding of an interesting work trying to address interesting questions. This work should not be published as is.

- Authors claim that the alignment between Image <-> Text tokens is what helps the model performing well on VQA tasks and hallucinations tasks. Something that is not tested is in which extend this is a cause (i.e performance on vision tasks is high thanks to that alignment), or a consequence (i.e alignment is high along performance on vision tasks thanks to the vision-language training). See "Questions" section of this review.

- Authors claim that an LLM can be seen as a residual stream with refinement blocks acting as steering blocks, and that this architecture is what helps the LLM to implicitly having an alignment between text and image tokens. It seems hard to support this claim without contrasting the work with another architecture. See "Questions" section of this review.

---

> ### Author Rebuttal · Authors · 2024-08-05
>
> We would like to thank the reviewer for the detailed feedback on our work. The comprehensive comments reflect a significant effort in reading our paper and providing feedback, which we highly appreciate. We recognize the reviewer's intention to help us further improve our paper and are grateful for their constructive input.
>
> ## **Weaknesses:**
>
> - **The presentation of the results and the overall quality**
>
> We appreciate the reviewer's suggestion to improve the readability of the figures. We have made significant efforts to ensure all numbers in the figures are as clear as possible (sometimes by zooming due to space constraints). To clarify, each figure in our submission is generated by a dedicated Python script that performs model  inference, calculated metrics, creates the figures, and saves them at a very high resolution. Initially, we received initial feedback from several readers about the large PDF file size and slow navigation of the PDF due to the high-resolution figures. Consequently, we compressed the figures to balance clarity and efficient navigation. After inspection, we did not find any figure that could make any claim ambiguous due to clarity.
>
> Regarding Figure 5, we acknowledge that readers may need to zoom in to read it. To address this, we have included a clearer version of Figure 5 in the appendix.
>
> Using TikZ packages would be an ideal solution for figure clarity, however, it requires significant effort to adapt our codes, rerun the experiments, and reproduce the figures with TikZ. We consider this for future papers.
>
> - **In many details, the work could have be improved (Claims and Typos).**
>
> Figure 1/12: We changed this to 50%. We didn’t put the 70% sparsity results in the paper because the Wanda pruning was not effective at this sparsity regime (see response to hn2C for more details)
>
> Prompt == Perceptual? Yes, we use them interchangeably as mentioned in L85 in Section 2. In our setup he prompt refers to tokens prepended before the textual ones. We unify the formatting to be $T$ and $P$ instead of T and P.
>
> - **Evaluation could be more complete.**
>
> The benchmarks that we use are typically correlated with other multimodal benchmarks. We agree that adding more recent ones will complete the work and we consider this for future iterations on the paper.
>
> - **Typos:**
>
> We tried as possible to follow the reviewer's suggestions. We update the paper accordingly and provide answers to raised questions:
>
> **What is the point of the right block named "Framework"?**
>
> To clarify that family of models that we analyze and to show that we focus on the LLM and look at the representations in the residual stream, between and inside the blocks.
>
> **The claim "Fig. 2 shows a clear narrow cone effect for textual and perceptual tokens." is not verifiable when looking at Figure 2.**
>
> We use the cosine similarity as an indicator of the cone effect. The unimodal tokens have high cosine similarity scores (p vs p) and (T vs T). This score is significantly larger than the cross-modal score (P vs T) or (P vs P in ST across different perceptual modalities). As high scores mean there is a cone effect, this means the narrow cones exist. As the cross-modal cosine similarity is small, this means the multimodal cones are distinct
>
> ## **Questions:**
>
> ### **Author claim about the alignment between Image <-> Text**
>
> - First, we would like to clarify that our paper discusses positive "correlation" and provides initial observations suggesting that the IMA score "could" serve as a proxy metric for task performance (framed as questions in the paragraph titles L238 and L243). We don’t claim that alignment leads to better performance or that this alignment should be maximized. Nonetheless this section occupies only half a page.
>
> - Despite being conservative in our claims, we have enough results to consider the IMA score a good candidate metric. We conducted a controlled experimental study by comparing different LLaVA variants (MT setup) and models trained for varying numbers of steps. These experiments showed indeed that the training improves both alignment and performance.
>
> - Additionally, and to support more our hypothesis, we add the following experiments (for the ST setup). The experiment consists of training LMMs with different image encoders; text-aligned (CLIP), self-supervised (MAE) and trained for imagenet (ViT),  We report the task performance and the IMA score (Fig.1 and Tab.1 in the uploaded pdf). Here we also observe that the alignment across correlates with performance: encoders with higher IMA scores (e.g. CLIP) also demonstrates better performance.
>
> - Rigorously studying this alignment across different setups and models is an interesting direction that we leave for future work. We thank the reviewer again for the very interesting suggestion.
>
> ### **Claim that an LLM can be seen as a residual stream with refinement blocks acting as steering blocks**
>
> - Contrasting the LLM architecture with other architectures would be necessary if we claimed that this is a unique property of LLM architecture, which we do not. We hypothesize that generalization to non-textual tokens is related to the architecture and provide observations to support this hypothesis.
>
> - The presence of similar architectural inductive biases (e.g., residuals or alternatives to attention modules) in other models does not change the main findings of the paper. Instead, it generalizes and solidifies the proposed hypothesis across a wider range of architectures.
>
> - While studying RNNs might be important, these architectures are not common in the multimodal community. Therefore, we prefer to focus on architectures that are widely adopted.
>
> - We thank the reviewer for this suggestion and leave pursuing this study in future work.
>
> ## **Limitations:**
>
> -  Please see answer to weakness 1
> -  Please see answer to question 1
> -  Please see answer to question 2

---

> > ### Comment · Reviewer_69kn · 2024-08-13
> >
> > I would like to thank the authors for their response. It is unclear why producing TikZ figures would require to rerun experiments as mentioned in the rebuttal. I assume the authors have kept all the outputs of their experiments in a readable format (csv, json, or even better wandb) for future consultation. My other concerns have been addressed. I keep my ratings unchanged.

---

### Official Review · Reviewer_dcgF · 2024-07-08

**Soundness:** 2
**Presentation:** 2
**Contribution:** 2
**Rating:** 5
**Confidence:** 2

**Summary:**

The paper explores how frozen Large Language Models generalize to multimodal inputs without the need for extensive re-training. It introduces the concept of Implicit Multimodal Alignment, which suggests that despite the distinct representations of perceptual and textual tokens, there exists a form of implicit alignment facilitated by the architecture of LLMs. The study leverages experimental setups across single-task and multitask environments to validate the IMA phenomenon and discusses its implications for computational efficiency and model performance.

**Strengths:**

1. The analysis spans various setups and modalities, providing a comprehensive look at how LLMs process multimodal inputs.
2. This study can quickly extend large semantic models to the multimodal domain, effectively reducing computational costs and enhancing the generalizability of LLMs.
3. The results are meaningful, proving the effectiveness of the method.

**Weaknesses:**

1. The concept of alignment within neural networks, although well-explored here, does not offer a groundbreaking methodological advance. The novelty lies more in the application context rather than in the development of new techniques or models.
2. The generalizability of the results is restricted to a subset of model architectures and sizes, potentially limiting the broader applicability of the findings.
3. The visualization experiments are not clear. Current figures are somewhat generic and do not adequately convey the unique aspects of the IMA effect.

**Questions:**

1. How does the IMA effect influence the overall performance of LLMs on standard multimodal tasks?  Are there performance trade-offs associated with the alignments observed?
2.  Are there indications that these findings could be applicable to other types of models or architectures not covered in the study?

**Limitations:**

1. The paper lacks implementation details.
2. The study focuses primarily on a specific range of LLMs. More domains and tasks should be included.

---

> ### Author Rebuttal · Authors · 2024-08-05
>
> We would like to thank the reviewer for the feedback. In the following, we answer the reviewer’s concerns.
> ## **Weaknesses:**
>
> - **The concept of alignment within neural networks, although well-explored here, does not offer a groundbreaking methodological advance. The novelty lies more in the application context rather than in the development of new techniques or models.**
>
> As far as we know, the significant gap between perceptual and textual tokens within LLMs and the ability of LLMs to generalize to non-textual tokens has not been studied before. While we respectfully disagree with the reviewer's categorization, this novelty does not diminish the importance of the messages conveyed by the paper.
>
> - **The generalizability of the results is restricted to a subset of model architectures and sizes, potentially limiting the broader applicability of the findings.**
>
> The primary motivation for our paper stems from the observation that simply connecting unimodal encoders to LLMs achieves unprecedented performance. This led us to consider two typical setups that use LLMs to tackle multimodal tasks. In the paper, we explore different LLMs (OPT, Vicuna v1.5), visual encoders (ViT, CLIP, MAE), and modalities (image, video, audio and text) as seen for example in Figs. 23, 26, 27, 28, 29, 32. We believe this covers a wide range of multimodal LLM approaches.
>
> Regarding the model size, 7B LLMs are currently widely used and have been shown to outperform many larger LLMs, making them an appealing choice for studying and understanding their capabilities.
>
> However, we agree that extending our study to other setups beyond LLMs is an interesting direction for future work, as stated in the discussion in Appendix B.
>
> - **The visualization experiments are not clear. Current figures are somewhat generic and do not adequately convey the unique aspects of the IMA effect.**
>
> We would appreciate more clarification on what the reviewer means by "not adequately convey the unique aspects of the IMA effect." This comment is too general, and it is unclear to us how to respond.
>
> ## **Questions:**
>
> ### **How does the IMA effect influence the overall performance of LLMs on standard multimodal tasks? Are there performance trade-offs associated with the alignments observed?**
>
> As we show in Figure 9, there is a positive correlation between the IMA score and performance on standard multimodal tasks; models with higher performance also have the highest IMA scores, and vice versa. Intuitively, better multimodal alignment leads to better understanding of different modalities. However, we do not claim that the two modalities must be fully aligned, and we leave this investigation for future work.
>
> ### **Are there indications that these findings could be applicable to other types of models or architectures not covered in the study?**
>
> Our experiments cover several LLMs with different encoders and connectors, trained on various multimodal datasets. We believe that these observations are generalizable to the broader category of multimodal LLMs (MLLMs). We leave conducting a more in-depth investigation of this for future work.
>
> ## **Limitations:**
> - **The paper lacks implementation details.**
>
> Due to space constraints, we included most of the details in the appendix. Could the reviewer point out details that should be considered in the main paper?
>
> - **The study focuses primarily on a specific range of LLMs. More domains and tasks should be included.**
>
> The current multimodal tasks we cover include question answering and text generation conditioned on images (i.e. image captioning), which cover many multimodal tasks (most of the tasks can be cast as question answering or instruction following). We address four modalities in our work. However, there are other domains of applications, such as 3D vision and speech, that we do not cover, but we believe that our findings should generalize to additional modalities and tasks.

---

> > ### Author Response · Authors · 2024-08-13
> >
> > Dear reviewer, we hope our clarifications can address your concerns, and we sincerely hope that you can reconsider our work in light of these clarifications. If you have any further comments, please do not hesitate to contact us. We greatly appreciate your contributions to the community.

---

### Official Review · Reviewer_hn2C · 2024-07-09

**Soundness:** 4
**Presentation:** 3
**Contribution:** 4
**Rating:** 9
**Confidence:** 4

**Summary:**

This paper conducts an in-depth study on the generalization capabilities of LLM when handling multimodal inputs without multimodal fine-tuning. It reveals the implicit multimodal alignment (IMA) effect between perceptual and textual tokens within LLMs and finds that this effect is closely related to the model architecture. The IMA effect contributes to enhanced task performance and reduced inference costs. Additionally, the paper proposes methods for model compression and reducing computational overhead, providing valuable insights for the future design and optimization of multimodal models.

**Strengths:**

Conducted an EXTENSIVE series of experiments to validate four hypotheses, providing insightful understanding of the mechanisms on how the MLLMs perceive multimodal information.

Based on the four findings, the paper offers relevant implications that explain the effects of IMA on tasks and hallucinations.

Proposes a novel approach to model compression by retaining a subnetwork.

Experiments are comprehensive and were conducted on various large language models, including OPT, Llama, and Vicuna.

**Weaknesses:**

The textual descriptions are somewhat difficult to understand, with limited explanation of the figures.

Insufficient explanation of the subnet part. It is unclear whether the similar performance after 50% sparsity is due to the effectiveness of the WANDA method or the extraction of the subnet.

**Questions:**

I do not fully understand line 247. How is the degree of hallucination measured? What is the relationship between cosine similarity and this measurement?

Out of personal curiosity, how long did these experiments take?

**Limitations:**

See weakness and question

---

> ### Author Rebuttal · Authors · 2024-08-05
>
> We thank the reviewer for very positive feedback about the paper and appreciate finding it novel, supported by an extensive series of experiments and insightful messages. This feedback encourages us to push further for similar interesting work. In the following we try to address all the reviewer's remaining concerns.
>
> ## Weaknesses:
>
> - **The textual descriptions are somewhat difficult to understand, with limited explanation of the figures.**
>
> We agree that some figures are not exhaustively explained in the captions. This is due to space limitation, we tried to explain rather in the text or in the appendix. We appreciate it if the reviewer can point to specific figures that need an elaborated explanation.
>
> - **Insufficient explanation of the subnet part. It is unclear whether the similar performance after 50% sparsity is due to the effectiveness of the WANDA method or the extraction of the subnet.**
>
>
> Based on our experiments, the significant drop in performance when exceeding 50% sparsity is attributed to Wanda. To support this, we provide additional scores when pruning the model with Wanda (OPT LLM with a CLIP encoder):
>
>
> | **Sparsity** | **COCO** | **VQAv2** | **GQA** | **OKVQA** |
> |--------------|----------|-----------|---------|-----------|
> | **0.5**  	| 126.81   | 55.28 	| 24.72   | 42.00 	|
> | **0.7**  	| 84.47	| 16.41 	| 10.70   | 1.80  	|
>
>
> These results demonstrate a substantial performance decrease with increased sparsity. It is also worth noting that similar observations have been reported in concurrent works [1] (Fig. 4a).
>
>
> [1] Sung, Yi-Lin, Jaehong Yoon, and Mohit Bansal. "Ecoflap: Efficient coarse-to-fine layer-wise pruning for vision-language models." arXiv preprint arXiv:2310.02998 (2023).
>
>
> ## Questions:
>
> ### **I do not fully understand line 247. How is the degree of hallucination measured? What is the relationship between cosine similarity and this measurement?**
>
> We follow standard practices for measuring multimodal hallucinations (specifically object hallucinations), which occur when the model describes or refers to objects not present in the input image.
>
> **Hallucination Metrics:**
> - COCO: On the COCO image captioning dataset, the model is asked to describe the images. We compute the CHAIR metrics based on the generated captions and the ground truth annotations of all objects in the image. If a caption contains non-existent objects, we classify it as a hallucinated caption. The CHAIRs score is the ratio of hallucinated captions to the total number of captions. Additionally, we calculate the ratio of hallucinated objects to the total number of objects across all captions, which is referred to as CHAIRi. A CHAIR score of 0 indicates no hallucinations. In the paper, we report (1 - CHAIR) × 100, so a higher score indicates fewer hallucinations.
>
> - POPE: This is a question-answering task involving questions about the existence of objects in images. The metric used is accuracy; the fewer the hallucinations, the higher the accuracy.
>
> We add these details to the appendix of the revised paper.
>
> **Link to image-text similarity**: In the paper, we hypothesize a link between the degree of cross-modal alignment and hallucinations. Our intuition is that for a textual model to understand the details in images, the image features should be well aligned to the textual domain. The better the alignment (measured here with cosine similarity) the better the model at performing reasoning about it. Misaligned representations can be considered loosely as out-of-distribution/modality samples which cause the model to be less confident and more likely to hallucinate.
>
> ### **Out of personal curiosity, how long did these experiments take?**
>
> The experiments involve both training models and analyzing them. Training LLaVA models typically takes less than one day on 8 GPUs, while ST models require a few hours, depending on the size of the dataset. Computing similarity at the token level also takes a few hours on a single GPU. However, due to the exploration of many different ideas, of which only a portion are included in the paper, the project consumes a significant amount of GPU hours. Generally, such a project can be conducted with an academic-level compute budget (8 GPUs).

---

### Official Review · Reviewer_vaQc · 2024-07-09

**Soundness:** 4
**Presentation:** 4
**Contribution:** 4
**Rating:** 7
**Confidence:** 4

**Summary:**

This work aims to understand multi-modality representation within MLLMs. It provides some interesting findings about how LLMs generalize to non-textual tokens and what helps LLMs to generalize to multimodal tokens. Additionally, several implications are proposed based on these findings.

**Strengths:**

1. The authors present many interesting findings about how LLMs generalize to non-textual tokens. These findings could help in understanding MLLMs and inspire future research.
2. Based on these findings, the authors propose several implications on performance, safety and efficiency.
3. The figures are well-illustrated and effectively support the findings.

**Weaknesses:**

**About experiments on α-SubNet.**

--Figure 12 appears incomplete. What does "Avg" mean in this table?

--The comparison between task-specific pruning approaches (Wanda) and the proposed task-agnostic approach is not comprehensive. For instance, what is the performance of Wanda sub-network pruning on COCO across other multimodal tasks? In other words, I'm curious about the generalization of the task-specific methods. Including these results would help in understanding the significance of task and modality-agnostic pruning methods.

--Additionally, the results in Table 1 of the Appendix should be incorporated into the main paper for better readability.

--Overall, I recommend reorganizing the experiments related to α-SubNet.

**Others**.

--line 1192 is unfinished.

**Questions:**

In the single-task (ST) setting (Fig. 2), the authors use unimodal encoders without text-aligned. Figure 2 illustrates "Multimodal cones: different narrow cones for different modalities." However, I wonder if this phenomenon is related to the unimodal encoders themselves—i.e., different modality features from unimodal encoders might have very different feature distributions, leading to different narrow cones in LLMs. Could the authors conduct new experiments using aligned encoders, such as ImageBind [1], as different modality encoders? This would help determine whether the different narrow cones in LLMs are due to different modalities or different encoders.

Ref:
1. ImageBind: One Embedding Space To Bind Them All. CVPR 2023

**Limitations:**

As stated by the authors, the generalization of the findings, to larger and more powerful models, with different architectures, including proprietary ones remains to be seen

---

> ### Author Rebuttal · Authors · 2024-08-05
>
> The authors would like to thank the reviewer for the positive feedback and appreciate finding the paper interesting, insightful and well presented. In the following we try to address all the remaining concerns:
>
> ## **Weaknesses**
>
> ### About experiments on α-SubNet.
>
> - **Clarification about Figure 12**: The "Avg" in Figure 12 represents the average score across all multimodal tasks (in this case, VQAv2 and COCO). We will update the figure to reflect this more clearly.
>
> - **Generalization of Task-Specific Methods**: This is an important point. We have included experiments in Figure 6, as well as Figures 26, 27, 28, and 29 in the appendix, that address the generalization of pruned masks/subnetworks to other tasks and modalities. All these results involve task-specific pruning methods (i.e., Wanda). For example, the performance of Wanda sub-network pruning on COCO across other multimodal tasks is as follows (Figure 6): 59.23 (vs 59.69) on VQAv2, 45.19 (vs 44.09) on GQA, and 29.30 (vs 30.89) on OKVQA which indicate very good generalization as the scores are comparable to pruning masks coming from the same datasets (indicated in parenthesis here). The generalization of task-specific pruning to other tasks was a main part of  the motivation for α-SubNet.
>
> - **Incorporation of Table 1 into the Main Paper**: We agree that Table 1 provides a stronger signal regarding the effectiveness of α-SubNet. However, due to space limitations, we included only a subset of the results. We update the revised paper by moving more benchmarks coming from additional modalities.
>
> ### Others
> **--line 1192 is unfinished.** Thanks for spotting this. This paragraph is redundant as it is detailed before in line in 1169. We remove this line in the revised paper.
>
> ## Questions:
>
> ### **Whether the different narrow cones in LLMs are due to different modalities or different encoders?**
> We can break this question to several sub questions:
>
> 1. Does the modality gap still exist for text-aligned encoders?
>
> Yes. To demonstrate that the modality gap (or different multimodal "cones") persists even for text-aligned features, we conduct a comprehensive comparison between different encoders (new Fig 1 in the uploaded pdf). This includes text-aligned encoders such as CLIP, unaligned encoders like MAE, and encoders trained for classification (e.g., ViT on imagenet). Our findings reveal that: (1) the modality gap persists even for encoders aligned with text; (2) CLIP encoders produce features that are more closely aligned with LLM textual features, while MAE produces the most misaligned features.
>
> 2. What causes different narrow cones in LLMs?
>
> Previous work [1]  has shown that even with the same model architecture and training data, varying the random seed can lead to differences in encoded representations (i.e., different narrow cones). Thus, differences among encoders, or the same encoders trained on different data or modalities, all contribute to having different narrow cones.
>
> 3. Does this affect the main paper message?
>
> Our goal is not to show that there is a gap between different multimodal encoders, but rather to highlight that this gap still exists within LLMs even after projecting all modalities into the same LLM input space. Despite this gap between perceptual and textual features, the LLM is still able to generalize effectively. Whether the multimodal aspect is the sole cause of this gap or if there are additional contributing factors, our main conclusions and messages remain valid.
>
>
> [1] Liang, Victor Weixin, et al. "Mind the gap: Understanding the modality gap in multi-modal contrastive representation learning." Advances in Neural Information Processing Systems 35 (2022): 17612-17625.

---

> > ### Comment · Reviewer_vaQc · 2024-08-12
> >
> > Thank the authors for the detailed response. All my concerns have been addressed. I'd like to keep my rating.

---

### Author Rebuttal · Authors · 2024-08-05

- We would like to thank the reviewers for the very detailed and positive feedback. We appreciate finding our work original (vaQc, hn2C, dcgF, 69kn), provide valuable and interesting insights (vaQc, hn2C, dcgF, 69kn), supported by extensive experimentation (hn2C) and well presented (vaQc). We find the feedback very encouraging with many interesting and relevant suggestions.

- Based on the reviewers suggestions, we made changes to the paper, some of them are mentioned in this rebuttal and the uploaded PDF, and others are integrated directly to the main paper. The changes mainly cover adding new experiments, clarification and improving the overall presentation (e.g. figures, typos). We detail the changes in our response to each reviewer.

- In the following we try to address all reviewers ' concerns.

---

### Decision · Program_Chairs · 2024-09-25

**Decision:**

Accept (poster)

**Comment:**

This is an original and exciting attempt to understand the underlying mechanics of how LLMs generalize beyond text tokens to process multimodal inputs. All the reviewers liked this work. For examples, Reviewer vaQc noted that "The authors present many interesting findings about how LLMs generalize to non-textual tokens. These findings could help in understanding MLLMs and inspire future research." Reviewer hn2C commented "Conducted an EXTENSIVE series of experiments to validate four hypotheses, providing insightful understanding of the mechanisms on how the MLLMs perceive multimodal information." Reviewer 69kn added that "This is a very original and in-depth work to understand the inside mechanisms appearing in MLLMs. Many/Most of the current published MLLMs are based on variations a pre-trained LLM + a pre-trained vision encoder with smaller VL connector in between. The insights from this study could be applied on many of those works, especially to better improve the visual faithfulness of those models."

Reviewer dcgF who seems most critical about the work, although still positive about the work, is expecting more "methodological advance" from the work, and points out other limitations of the work such as findings specific to model architectures. I think the authors addressed these concerns well in the rebuttal.

In conclusion, this is an exciting work to advance our understanding about LLMs, which should be of highly interest to NeurIPS audience.